EMBO
Molecular Medicine

# Aberrant fragmentomic features of circulating cell-free mitochondrial DNA as novel biomarkers for multi-cancer detection

Yang Liu[1,2,4], Fan Peng[1,4], Siyuan Wang[1,4], Huanmin Jiao[1], Miao Dang[1], Kaixiang Zhou[1], Wenjie Guo [ID][1], Shanshan Guo[1], Huanqin Zhang[1], Wenjie Song[3] & Jinliang Xing [ID][1✉]

## Abstract

**Fragmentomic features of circulating cell free mitochondrial DNA (ccf-mtDNA) including fragmentation profile, 5′ end preference and motif diversity are poorly understood. Here, we generated ccf-mtDNA sequencing data of 1607 plasma samples using capture-based next generation sequencing. We firstly found that fragmentomic features of ccf-mtDNA were remarkably different from those of circulating cell free nuclear DNA. Furthermore, region-specific fragmentomic features of ccf-mtDNA were observed, which was associated with protein binding, base composition and special structure of mitochondrial DNA. When comparing to non-cancer controls, six types of cancer patients exhibited aberrant fragmentomic features. Then, cancer detection models were built based on the fragmentomic features. Both internal and external validation cohorts demonstrated the excellent capacity of our model in distinguishing cancer patients from non-cancer control, with all area under curve higher than 0.9322. The overall accuracy of tissue-of-origin was 89.24% and 87.92% for six cancer types in two validation cohort, respectively. Altogether, our study comprehensively describes cancer-specific fragmentomic features of ccf-mtDNA and provides a proof-of-principle for the ccf-mtDNA fragmentomics-based multi-cancer detection and tissue-of-origin classification.**

**Keywords** Fragmentomics; Circulating Cell-Free Mitochondrial DNA; Multi-Cancer Detection; Tissue-of-Origin Classification; Biomarker
**Subject Categories** Biomarkers; Cancer

## Introduction

The poor prognosis of numerous patients with cancer are mainly caused by late diagnosis when therapeutic intervention is less effective, especially in developing countries. Unfortunately, clinically proven biomarkers for cancer detection face many challenges, including low sensitivity and specificity, complicated operations, and data analyses. Liquid biopsy, a powerful complement to traditional methods, has made significant breakthroughs in the clinical practices of cancer detection (Nikanjam et al, 2022). Currently, research is focused on the detection of circulating tumor DNA (ctDNA), including copy number variations, somatic mutations, and methylation (Cescon et al, 2020). Profiling copy number variations and somatic mutations throughout the whole genome has provided feasible methods for cancer detection. However, their sensitivities remain suboptimal, especially in early-stage cancers (Benesova et al, 2013; Tao et al, 2020; Wang et al, 2021). Furthermore, although a series of methods have been developed, the process of ctDNA methylation analysis remains complex because of the bisulfite treatment (Sestakova et al, 2019; Kerachian et al, 2021). And endonuclease digestion- or affinity enrichment-based methods for ctDNA methylation detecting are prone to sources of bias (Li and Tollefsbol, 2021).

Recently, cell-free DNA (cfDNA) fragmentomic features, which encompass fragment size, end motifs, breakpoint motifs, and nucleosome footprints, have demonstrated the potential of cfDNA fragmentomics in detecting cancers, including lung cancer (Mathios et al, 2021), colorectal cancer (Ma et al, 2021), hepatocellular carcinoma (Jin et al, 2021). Cristiano et al reported that the genome-wide fragmentation profile of cfDNA differed between patients with cancer and healthy individuals (Cristiano et al, 2019). Bao et al recently constructed an early detection model for multiple cancers based on cfDNA fragmentomic features obtained from whole-genome sequencing (WGS) data (Bao et al, 2022). However, whether the repeatability of the cfDNA fragmentation profile is high enough needs to be further validated given the low coverage of sequencing, which may limit their translational application.

Inside the cells, except for the nuclei, the mitochondria have their own double-stranded circular genomic DNA with a length of 16.6 kb and a high copy number, which also contributes to the total content of circulating cfDNA through cell death or exosome-mediated release and holds great promise for cancer detection (Li et al, 2016; Weerts et al, 2018; Gambardella et al, 2019; Yin et al,

[1]State Key Laboratory of Holistic Integrative Management of Gastrointestinal Cancers and Department of Physiology and Pathophysiology, Fourth Military Medical University, Xi'an, China. [2]Department of Clinical Diagnosis, Tangdu Hospital, Fourth Military Medical University, Xi'an, China. [3]Department of Hepatobiliary Surgery, Xijing Hospital, Fourth Military Medical University, Xi'an, China. [4]These authors contributed equally: Yang Liu, Fan Peng, Siyuan Wang. ✉E-mail: xingjl@fmmu.edu.cn

2019; Johnson et al, 2022; van der Pol et al, 2023). Tumor-derived circulating cell-free mtDNA (ccf-mtDNA) is present in patients with many types of cancer (He et al, 2010; Newell et al, 2018; Perdas et al, 2019; Bulgakova et al, 2021). Moreover, several studies have revealed a remarkably smaller fragment size of ccf-mtDNA when compared to cell-free nuclear DNA (ccf-nDNA), and have shown significantly increased copy numbers in hepatocellular carcinoma (Jiang et al, 2015), lung cancer (Bulgakova et al, 2022), breast cancer (Pasha et al, 2019) and colorectal cancer (Xu et al, 2021). Furthermore, it is more sensitive to monitor tumor burden by detecting tumor-derived mtDNA than circulating tumor nDNA analysis (Mair et al, 2019). Our previous study demonstrated that the end motifs and mutation profiles of urine cf-mtDNA have diagnostic potential in various cancer patients. Furthermore, given the small size and high copy number of the mitochondrial genome, capture-based mtDNA sequencing could provide a more affordable alternative method that enables high coverage depth to improve the sensitivity and repeatability of cancer detection. However, few studies have reported the fragmentomic features and clinical applications of ccf-mtDNA.

Here, we comprehensively described the cancer-specific fragmentomic features of ccf-mtDNA using sequencing data from 1607 plasma samples and demonstrated the clinical value of ccf-mtDNA fragmentomic features as novel biomarkers for multi-cancer detection and tissue-of-origin classification, with improved performance over existing approaches.

# Results

## ccf-mtDNA exhibits different fragmentomic features from ccf-nDNA

Given the differences in size, structure, copy number, and location between nDNA and mtDNA, we hypothesized that ccf-mtDNA would exhibit fragmentomic features that are different from those of ccf-nDNA. To evaluate the repeatability of our detecting procedure for plasma ccf-mtDNA fragmentomic features, 10 healthy individuals were randomly selected, and the cfDNA sample was subject to three times library construction and WGS (Appendix Fig. S1a). No significant difference was observed in the detection of ccf-mtDNA fragmentomic features among the three replicates, which also suggested no notable bias among different test batches (all $P > 0.05$, Appendix Fig. S1b). Then, WGS data were obtained from 30 randomly selected plasma samples from healthy individuals for comparing the fragmentomic features of ccf-mtDNA (average sequencing coverage = 71.91 X) and ccf-nDNA (average sequencing coverage = 8.93 X). In addition, our results indicated the stable fragmentomic features when the depth of ccf-nDNA and ccf-mtDNA data were over 1 X and 50 X, respectively (Appendix Fig. S2). Compared to ccf-nDNA, ccf-mtDNA exhibited a different fragment distribution, with a significantly shorter median fragment size (128 bp vs. 166 bp, $P < 0.0001$; Fig. 1A,B) but a larger coefficient of variation (CV) of median fragment size (0.1294 vs. 0.0119, Fig. 1C). The 5′ end base preference and 5′ end motif diversity are other important fragmentomic features, which were preserved in end repair process during sequencing library construction. As shown in Fig. 1D, ccf-mtDNA exhibited a significantly lower preference for 5′ C-end and 5′ T-end but a significantly higher

preference for 5′ G-end and 5′ A-end compared to ccf-nDNA (all $P < 0.05$). In addition, ccf-mtDNA had a significantly larger CV of 5′ end base preference when compared with ccf-nDNA (Fig. 1E). Moreover, our data showed that ccf-mtDNA had a higher 5′ end motif diversity score (MDS) ($P < 0.001$; Fig. 1F) and CV of MDS (0.0058 vs. 0.0009, Fig. 1G) when compared with ccf-nDNA. Similar results were observed in patients with cancer and other non-cancerous diseases (Appendix Fig. S3). Our results indicate that ccf-mtDNA exhibits fragmentomic features different from those of ccf-nDNA, which is a universal principle across various populations.

## Capture-based NGS reveals the characteristic fragmentation profile of ccf-mtDNA

Considering the non-random process of ccf-mtDNA cleavage and fragmentation, we further explored whether a characteristic fragmentation profile of ccf-mtDNA exists at a single-base resolution based on deep capture-based mtDNA sequencing (mtDNA Cap-seq) established in our lab as an alternative cost-effective approach (Liu et al, 2021), which detected highly consistent fragmentomic features of ccf-mtDNA with WGS (Appendix Fig. S4). Remarkably, a characteristic ccf-mtDNA fragmentation profile, depicted by the fragment size distribution (FSD) score of each mtDNA site, was observed in plasma samples from 30 healthy control (HC), with many obvious region-specific peaks across the mitochondrial genome, demonstrating a region-dependent distribution of fragment length, which was mainly determined by nuclease digestion (Fig. 2A). As shown in Appendix Fig. S5, a very similar fragmentation profile was observed between the WGS and mtDNA Cap-seq data, with a high correlation in the FSD score. Moreover, we demonstrated that placement of whole blood within 3 h at 4 °C had no discernible influence on ccf-mtDNA fragmentation profiles, whereas placement over 12 h at 4 °C led to a significant change of ccf-mtDNA fragmentation profiles (Appendix Fig. S6). We also demonstrated the accurate detection of ccf-mtDNA fragmentation profiles at more than 500 X coverage by downsampling mtDNA Cap-seq data to different sequencing coverages (Appendix Fig. S7). We also observed no apparent influence of age or sex on the ccf-mtDNA fragmentation profiles (Appendix Fig. S8). We hypothesized that protein binding would explain the fragmentation profiles of ccf-mtDNA and that long fragments might be abundant in protein-rich regions, resulting in low standardized mtDNA depth in the assay for transposase-accessible chromatin with sequencing (ATAC-seq) data. To test our hypothesis, we determined the standardized mtDNA depth in public single-cell ATAC-seq data optimized for mitochondrial DNA (mtscATAC-seq) (Lareau et al, 2021) and analyzed its correlation with the FSD score of ccf-mtDNA at each mtDNA site. As expected, an overall negative correlation with Spearman r of −03933 was found between the standardized mtDNA depth and the fragmentation profiles of ccf-mtDNA (Fig. 2B; Appendix Fig. S9). Four selected regions were enlarged in Fig. 2B, with Spearman correlation r of −0.7359, −0.7778, −0.9284, and −0.7922, respectively. Our data illustrate the characteristic fragmentation profile of ccf-mtDNA at single-base resolution, which may be mainly caused by protein binding to the mitochondrial genome.

Considering that fragmentation profile, 5′ end base preference, and 5′ end MDS are closely related to the ccf-mtDNA cleavage process, these features could be independent events or not. We found no notable difference in 5′ end base preference between short

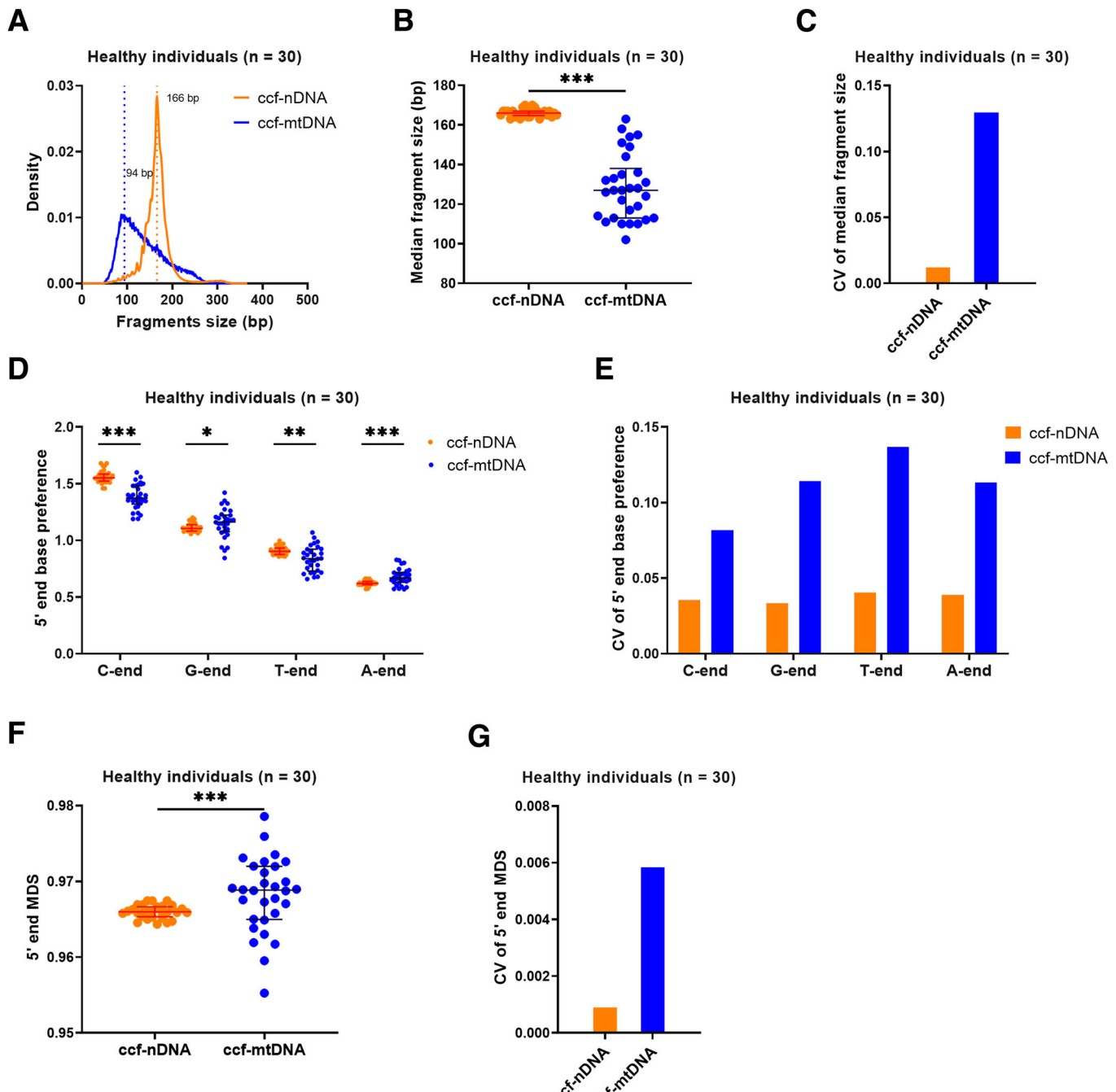

**Figure 1. Comparative analyses of fragmentomic features between ccf-nDNA and ccf-mtDNA based on WGS data from 30 healthy individuals.**

(A–G) Density of fragment size (A), median fragment size (B) (*P* value: <0.0001) and their CV (C), 5′ end base preference (D) (*P* values: <0.0001; 0.0411; 0.0022; 0.0004) and their CV (E), 5′ end MDS (F) (*P* value: 0.0009) and their CV (G) between ccf-nDNA and ccf-mtDNA. 5′ end base preference was calculated based on the proportion of the 5′ end base and the base composition of the mitochondria reference genome. 5′ end MDS was calculated based on the proportion of 254 4-mer end motifs. MDS motif diversity score, CV coefficient of variation; *P < 0.05; **P < 0.01; ***P < 0.001 (Wilcoxon rank-sum test). In (B, D, F), the center line indicates the median, lower, and upper hinges represent the 25th and 75th percentiles, respectively.

and long fragments, as determined by the median fragment size for each sample (*P* > 0.05; Appendix Fig. S10a). In contrast, short fragments showed significantly larger 5′ end MDS than long fragments (*P* < 0.05; Appendix Fig. S10b). Our results imply a more complex cleavage of short fragments.

**Fragmentomic features of ccf-mtDNA are highly related to functional regions of the mitochondrial genome**

Next, we compared the fragmentomic features of ccf-mtDNA in the four major functional regions of the mitochondrial genome. As

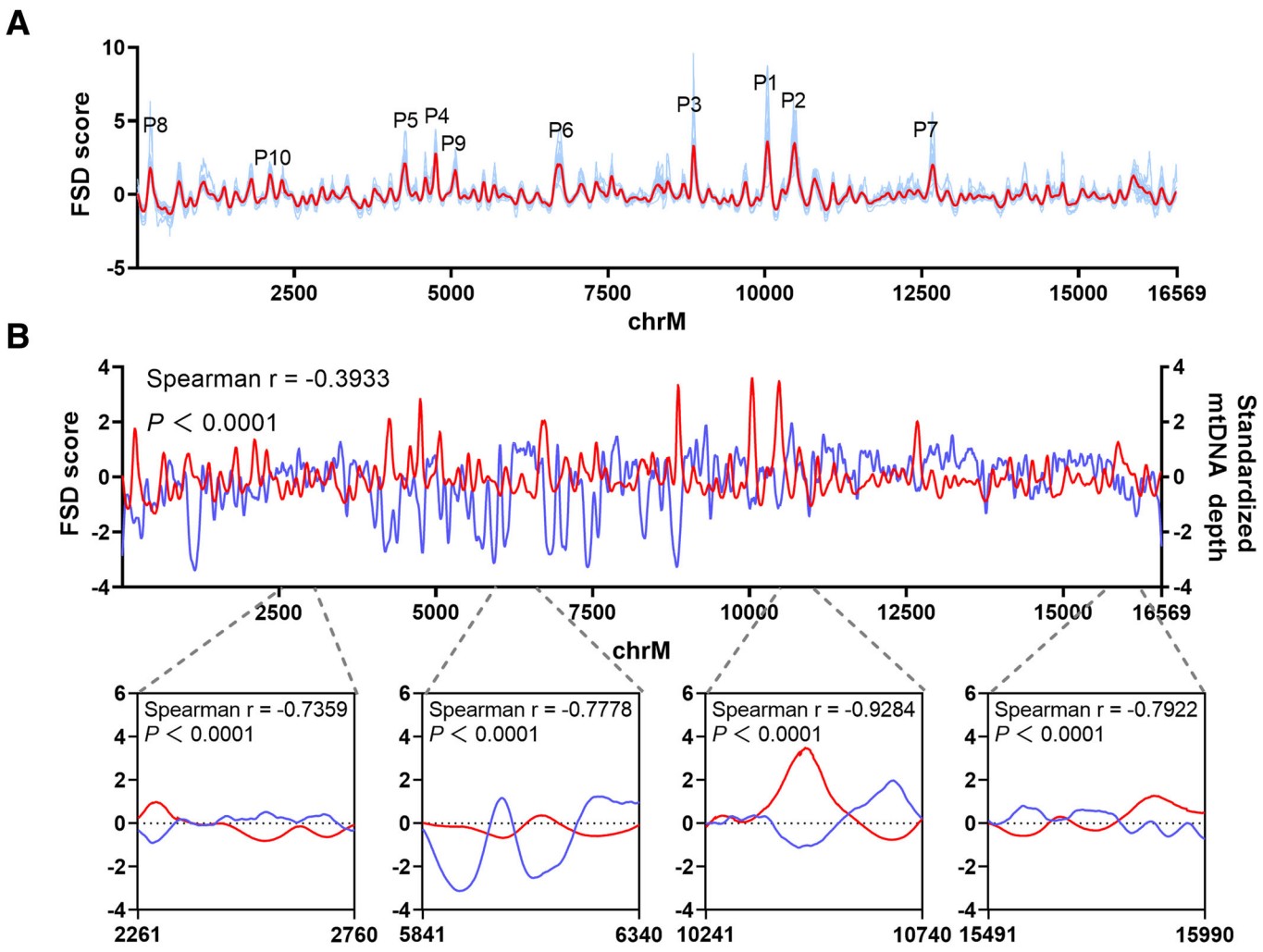

**Figure 2. Fragmentation profiles of ccf-mtDNA and correlation with mtDNA coverage depth.**

(A) Fragmentation profiles of ccf-mtDNA across the whole mitochondrial genome in capture-based mtDNA sequencing data of 30 HCs. The fragmentation profile of 30 HCs based on the median FSD score at each mtDNA site, which was scaled into the z-score was indicated in red and fragmentation profile of each HC was shown in blue. P1 to P10 represent the top ten peaks in peak height in the ccf-mtDNA fragmentation profile of median HC. (B) Correlation of the ccf-mtDNA fragmentation profile of median HC (red) with the medium standardized mtDNA depth (blue) scaled into the z-score from mtscATAC-seq data of 30 lymphoblastoid cells (P values: <.0001; <0.0001; <0.0001; <0.0001). HC healthy control, FSD fragment size distribution, mtscATAC-seq mitochondrial single-cell ATAC sequencing. Spearman's rank correlation coefficient was used to measure the associations between the two groups.

shown in Fig. 3A, the average FSD score of ccf-mtDNA in the D-loop, rRNA, mRNA, and tRNA regions ranked from low to high in the plasma samples from 300 HCs, with no significant difference between the rRNA and mRNA regions ($P > 0.05$). Similar results were obtained for the fragment size analysis (Fig. 3B,C). Our analysis further showed that the fragments covering four different regions exhibited a remarkable difference of 5′ end base preference ($P < 0.05$; Fig. 3D). The D-loop region had the smallest MDS value, followed by the tRNA region, whereas the mRNA and rRNA regions had the largest MDS values (0.9307 vs. 0.9440 vs. 0.9667 vs. 0.9633, $P < 0.05$; Fig. 3E). Similar differences in the fragmentomic features of ccf-mtDNA among the four regions were also observed in plasma samples from patients with various diseases, including malignant tumors (MT), benign tumors (BT), and inflammatory

diseases (INF) (Appendix Fig. S11). These results demonstrate that the fragmentomic features of ccf-mtDNA are highly related to the functional regions of the mtDNA.

## Chromatin conformation of mtDNA contributes to the functional region-related fragmentomic features of ccf-mtDNA

Furthermore, we determined the standardized coverage depth of the four mtDNA regions in the public mtscATAC-seq data to assess the possible influence of binding proteins on ccf-mtDNA fragmentation. As expected, the D-loop region exhibited the highest standardized coverage depth (all $P < 0.01$), whereas the tRNA region had the lowest depth (all $P < 0.001$), with no

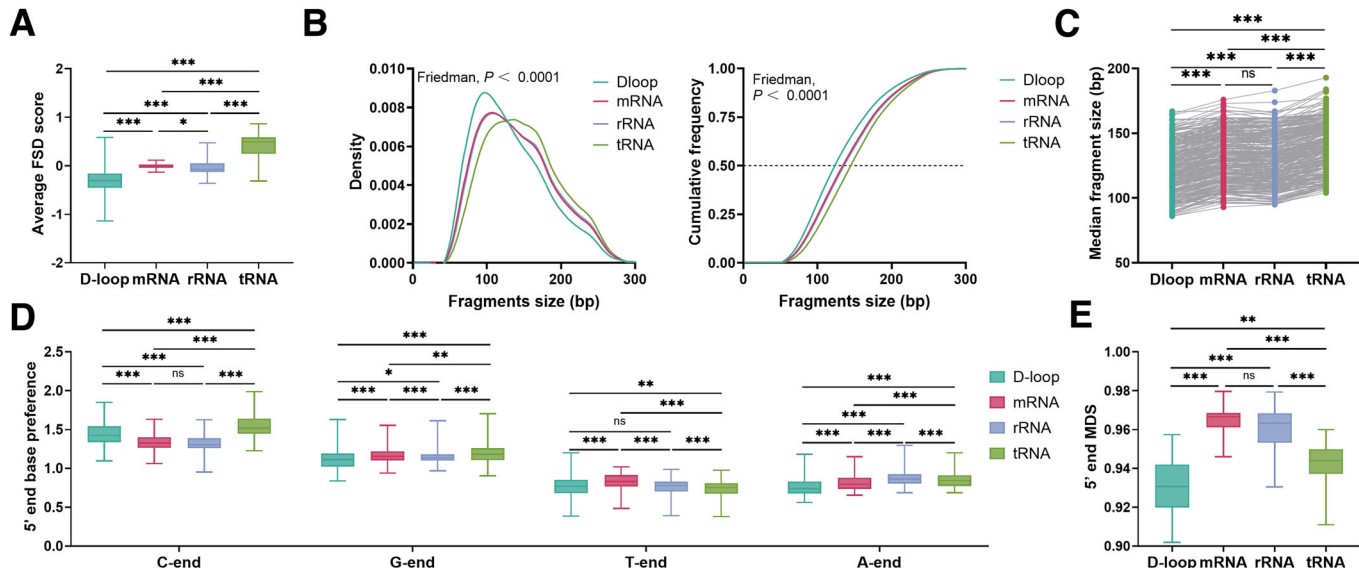

**Figure 3.  Different fragmentomic features of ccf-mtDNA among four functional regions of mitochondrial genome.**

(A–E) Comparison of average FSD score (A) (*P* values: <0.0001; <0.0001; <0.0001; 0.0437; <0.0001; <0.0001), size distribution and the cumulative frequency of the fragments (B) (*P* values: <0.0001; <0.0001), the median fragment size (C) (*P* values: <0.0001; <0.0001; <0.0001; <0.0001; <0.0001), 5′ end base preference (D) (*P* values: C-end: <0.0001; <0.0001; <0.0001; <0.0001; <0.0001, G-end: <0.0001; 0.0444; <0.0001; 0.0002; 0.0058; <0.0001, T-end: <0.0001; 0.0032; <0.0001; <0.0001; <0.0001, A-end: <0.0001; <0.0001; <0.0001; <0.0001; <0.0001; 0.0002) and 5′ end MDS (E) (*P* values: <0.0001; <0.0001; 0.0057; <0.0001; <0.0001) of ccf-mtDNA among D-loop region, protein-coding (mRNA) regions, rRNA and tRNA regions based on mtDNA Cap-seq data from plasma samples of 300 healthy individuals. The average FSD score for each individual was calculated based on all sites in a given functional region. FSD, fragment size distribution; ns not significant; *P < 0.05; **P < 0.01; ***P < 0.001 (Kruskal–Wallis *H*-test and Dunn's multiple comparisons test). The Friedman test was used to compare the difference in fragment size distribution. In (A, D, E), boxes represent the 25th–75th percentiles, the center line indicates the median, whiskers extend to the maximum and minimum values within 1.5× interquartile range.

significant difference between the mRNA and rRNA regions (*P* > 0.05) (Fig. 4A). Considering previous reports that DNA regions with low G and C base contents are more likely to bind proteins (Dekker, 2007), we calculated the base composition of the four functional regions in the revised Cambridge mtDNA reference sequence (rCRS). As shown in Fig. 4B, a significant difference in G&C content was observed among the D-loop, mRNA, rRNA, and tRNA regions, with the highest in the D-loop region (46.79%) and the lowest in the tRNA regions (36.63%). No significant difference in G&C content was observed between the mRNA (44.68%) and rRNA (44.88%). Further analyses of mtDNA Cap-seq data from the plasma samples of 300 HCs indicated that the A&T base sites of ccf-mtDNA had significantly higher FSD scores than the G&C base sites (*P* < 0.001; Fig. 4C). In contrast, public mtscATAC-seq data showed that the A&T base sites of mtDNA had a significantly lower standardized coverage depth than the G&C base sites (*P* < 0.001; Fig. 4D). The mitochondrial D-loop is a triple-stranded region formed by the stable incorporation of a third short DNA strand, known as 7S DNA (Nicholls and Minczuk, 2014). Therefore, we explored the fragmentomic features of non-7S and 7S DNA regions in the D-loop. Our results showed that the 7S DNA region had significantly shorter fragments, stronger 5′ C-end base preference, weaker 5′ G-end and 5′ T-end base preference, lower average FSD score, and higher standardized mtDNA depth than non-7S DNA region (all *P* < 0.05; Fig. 4E–H). Altogether, our findings suggest that the chromatin conformation, including protein binding, base composition, and the special structure of mtDNA, may greatly contribute to region-specific ccf-mtDNA fragmentation.

## Aberrant fragmentation profiles of ccf-mtDNA in six cancer types

To elucidate the aberrant fragmentation profiles of ccf-mtDNA in patients with cancer, mtDNA Cap-seq with >500 X coverage was performed on the plasma samples of 877 patients with MT, 140 patients with BT, 290 patients with INF, and 300 HCs. Remarkably, the MT group had highly variable profiles with a significantly decreased correlation with the median fragmentation profile of 30 HCs aforementioned when comparing with HC, BT and INF groups (all *P* < 0.001, Fig. 5A,B). And we also found a weak decreased correlation in BT and INF groups when comparing with HC (all *P* < 0.05, Fig. 5A,B). We then calculated the peak number of fragmentation profiles in all groups and defined new peaks based on site information by comparing with the median HC profile. As shown in Fig. 5C, the MT group had the largest number of new peaks compared to the other three groups, including HC, INF, and BT (81 vs. 51 *vs.* 68 vs. 60, *P* < 0.001). A median of 42.5 new peaks only present in more than 20% of patients with the six cancer types, but not in HC, patients with BT, and patients with INF are indicated by the red line in Fig. 5D, implying potential cancer-specific alterations.

Further analyses indicated a difference of 5′ A-end, T-end, and C-end but not G-end base preference among all groups (Appendix Fig. S12a–d). An increased 5′ end MDS and mtDNA copy number (CN) was found in patients with MT compared to the other three groups (*P* < 0.05; Appendix Fig. S12e,f). Given that the FSD score was calculated based on fragment size and coverage, we investigated

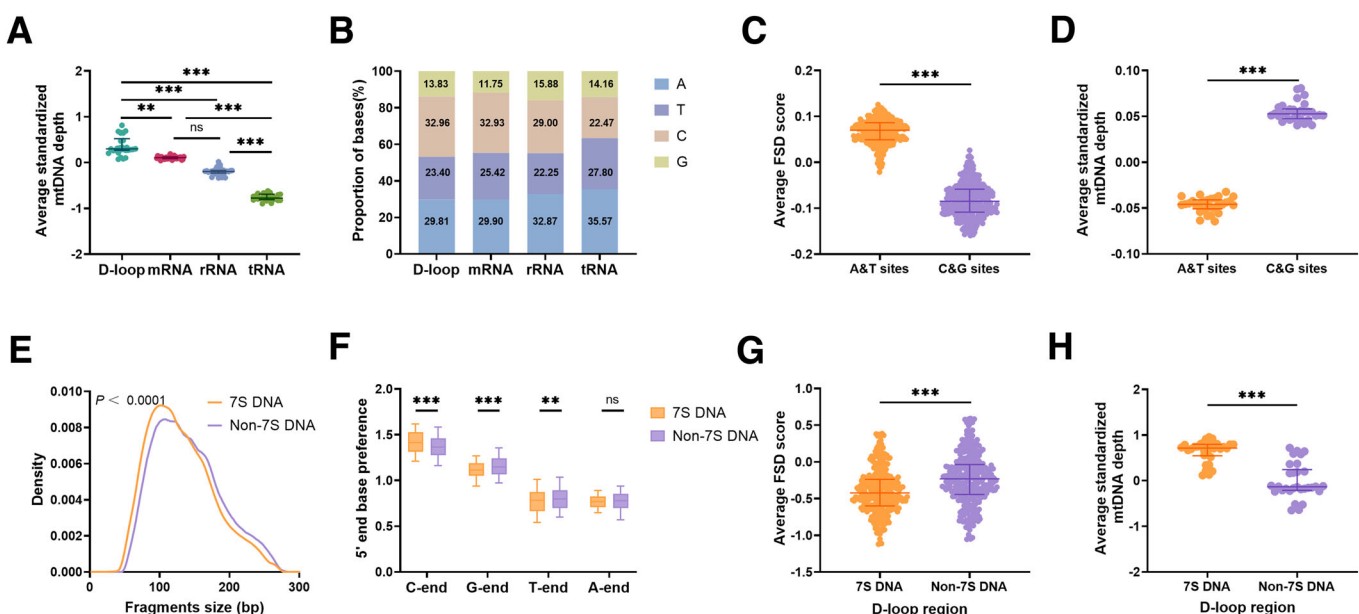

**Figure 4. Chromatin conformation of mtDNA contributes to the functional region-related fragmentomic features of ccf-mtDNA.**

(A) Comparison of average standardized mtDNA depth among D-loop, mRNA, rRNA, and tRNA regions (P values: 0.0091; <0.0001; <0.0001; <0.0001; <0.0001). (B) The proportion of four bases in D-loop, mRNA, rRNA, and tRNA regions of a mitochondrial reference genome (revised Cambridge reference sequence, rCRS). (C, D) Average FSD score and standardized mtDNA depth of A&T base sites and G&C base sites across the whole mtDNA (P values: C: <0.0001; D: <0.0001). (E–H) Comparison of the fragment size distribution, 5′ end base preference, average FSD score of ccf-mtDNA, and average standardized mtDNA depth between 7S DNA region and Non-7S DNA region of D-loop region (P values: E: <0.0001; F: <0.0001; <0.0001; 0.0012; G: <0.0001; H: <0.0001). The average FSD score for each individual was calculated based on all sites in a given functional region. Fragmentomic features of ccf-mtDNA were determined in mtDNA Cap-seq data from plasma samples of 300 healthy individuals, while average standardized mtDNA depth for all sites in a given region was determined based on mitochondrial single-cell ATAC-seq data from 30 lymphoblastoid cells. FSD, fragment size distribution; MDS, motif diversity score; ns not significant; **P < 0.01; ***P < 0.001 (Wilcoxon rank-sum test for comparing two sets of data, and Kruskal–Wallis H-test and Dunn's multiple comparisons test for comparing multiple sets of data). The Wilcoxon signed-rank test was used to compare the difference in fragment size distribution. In (A, C, D, G, H), the center line indicates the median, lower and upper hinges represent the 25th and 75th percentiles, respectively. In (F), boxes represent the 25th–75th percentiles, the center line indicates the median, whiskers extend to the maximum and minimum values within 1.5× interquartile range.

whether ccf-mtDNA fragment sizes differed across disease types. As shown in Fig. 5E, patients with MT exhibited significantly shorter ccf-mtDNA fragments than those in the other three groups (P < 0.001). Furthermore, we explored a possible explanation for the shorter fragments in patients with MT. We detected 138 tumor-specific mtDNA mutations and 1192 germline variants in ccf-mtDNA from plasma samples, based on mtDNA Cap-seq data from 40 patients with hepatocellular carcinoma (Table EV1). Our data revealed that ccf-mtDNA fragments with tumor-specific mutations were significantly shorter than wild-type ccf-mtDNA fragments (P < 0.001; Fig. 5F), and had a lower 5′ C-end base preference, higher 5′ A-end base preference, and larger MDS (P < 0.001; Appendix Fig. S13a,b). No differences in median length, 5′ end base preference, and 5′ end MDS were observed between wild-type ccf-mtDNA fragments and those with germline variants (all P > 0.05; Fig. 5G; Appendix Fig. S13c,d).

## Fragmentomic features of ccf-mtDNA as novel biomarkers for cancer detection and tissue-of-origin classification

We further evaluated potential values of ccf-mtDNA fragmentomic features in cancer detection and tissue-of-origin classification by developing a novel approach named "mtDNA evaluation of fragmentomics for cancer investigation" (MEFI). As shown in

Appendix Fig. S14, the random forest algorithm was used to construct cancer detection models based on different fragmentomic features in the three different training cohorts. The MEFI score was obtained based on a tenfold cross-validation procedure to classify individuals as patients with MT, BT, INF, or healthy. Receiver operating characteristic (ROC) curves showed the high performance of the three cancer detection models in the training cohort, with area under the curve (AUC) of 0.9961, 0.9585, and 0.9774 for MT vs. HC, MT vs. INF, and MT vs. BT, respectively (Appendix Fig. S15). Furthermore, our data showed that the MEFI score reached an AUC of 0.9845 (95%CI, 0.9736–0.9954) in differentiating patients with MT from HC in the internal validation cohort, yielding a sensitivity of 90.31% (95%CI, 86.09–93.35%) and specificity of 90.67% (95%CI, 81.87–95.41%) (Fig. 6A; Table EV2). High performance was also observed for the MEFI score in distinguishing MT from BT and from INF, with AUC of 0.9446 and 0.9597, respectively. Similar results were found in the external validation cohort with AUC of 0.9796, 0.9322, and 0.9531 in differentiating patients with MT from HC, from BT and from INF, respectively (Fig. 6B; Table EV2). Moreover, considering the clinical value of the MEFI score in distinguishing the six types of cancer (non-small cell lung cancer (NSCLC), hepatocellular carcinoma (HCC), colorectal cancer (CRC), serous ovarian cancer (SOC), breast cancer (BC), and clear cell renal cell carcinoma (ccRCC)) separately, the model achieved all AUC higher than

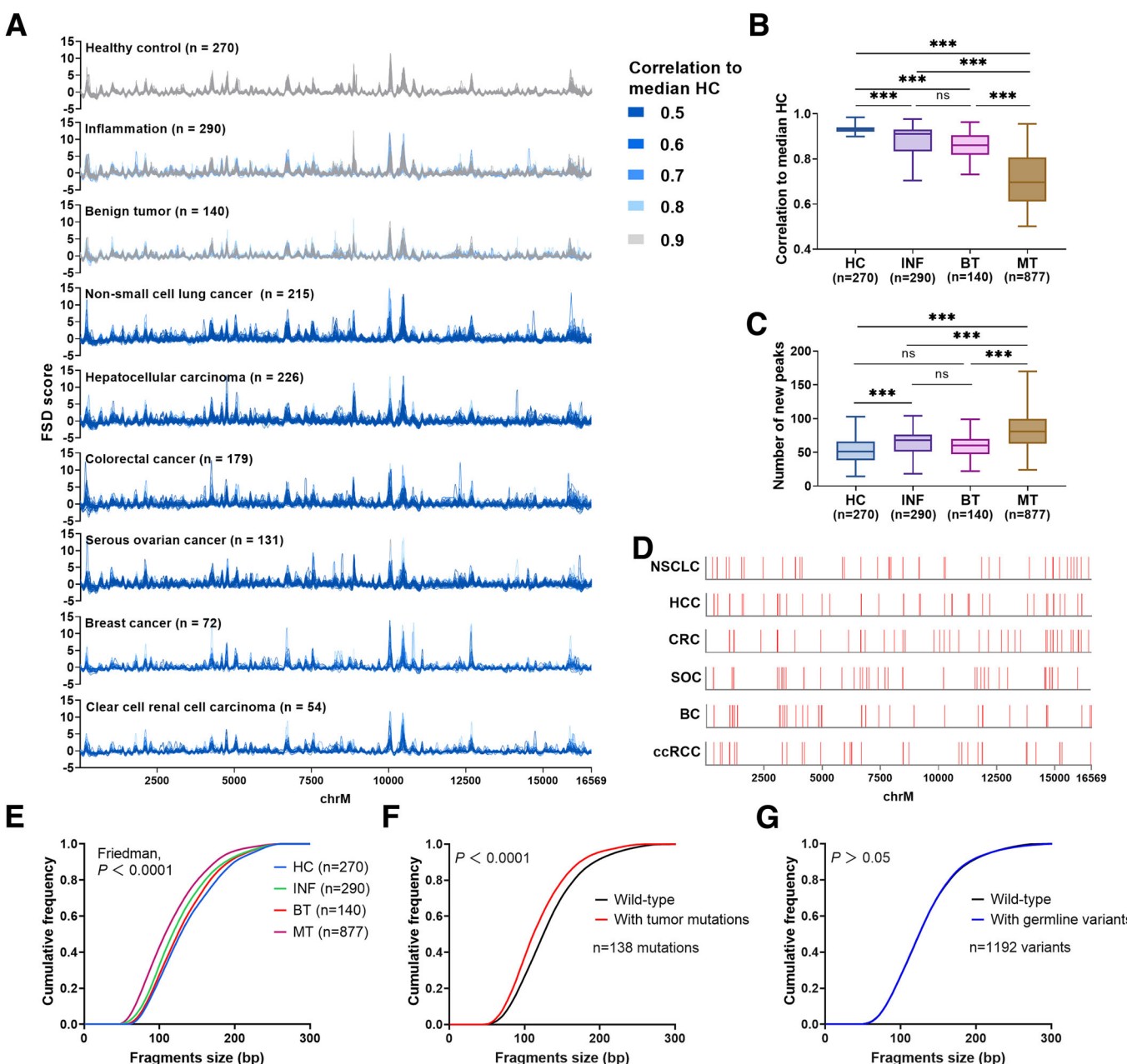

**Figure 5.  The ccf-mtDNA fragmentation profiles in HCs and patients with malignant tumor, benign tumor, and inflammatory diseases (MT, BT, and INF).**

(A) The ccf-mtDNA fragmentation profiles from capture-based mtDNA sequencing data for HCs and patients with different diseases. Individual profile for all HCs and patients with MT, BT, and INF was colored based on their correlation to the ccf-mtDNA fragmentation profile of median HC. (B, C) Comparison of the correlation to ccf-mtDNA fragmentation profile of median HC and the number of new peaks among four groups (P values: B: <0.0001; <0.0001; <0.0001; <0.0001; <0.0001, C: <0.0001; <0.0001; <0.0001). (D) Sites on the mitochondrial genome were indicated in red, where more than 20% of the MT patients had a new peak compared to the ccf-mtDNA fragmentation profile of median HC. (E) The cumulative frequency of ccf-mtDNA fragment size among four groups (P value: <0.0001). (F) Comparison of the cumulative frequency of ccf-mtDNA fragment size between the fragments with any 138 tumor-derived mutations (red) and corresponding wild-type fragments (black) in 40 patients with HCC (P value: <0.0001). (G) Comparison of the cumulative frequency of ccf-mtDNA fragment size between the fragments with any 1192 germline variants (blue) and corresponding wild-type fragments (black) in 40 patients with HCC. HC healthy control, MT malignant tumor, BT benign tumor, INF inflammation, HCC hepatocellular carcinoma, ns not significant; ***P < 0.001 (Kruskal–Wallis H-test and Dunn's multiple comparisons test). The Friedman test was used to compare the difference in fragment size distribution among the four groups, and the Wilcoxon signed-rank test was used for two groups. In (B, C), boxes represent the 25th–75th percentiles, center line indicates the median, whiskers extend to the maximum and minimum values within 1.5× interquartile range.

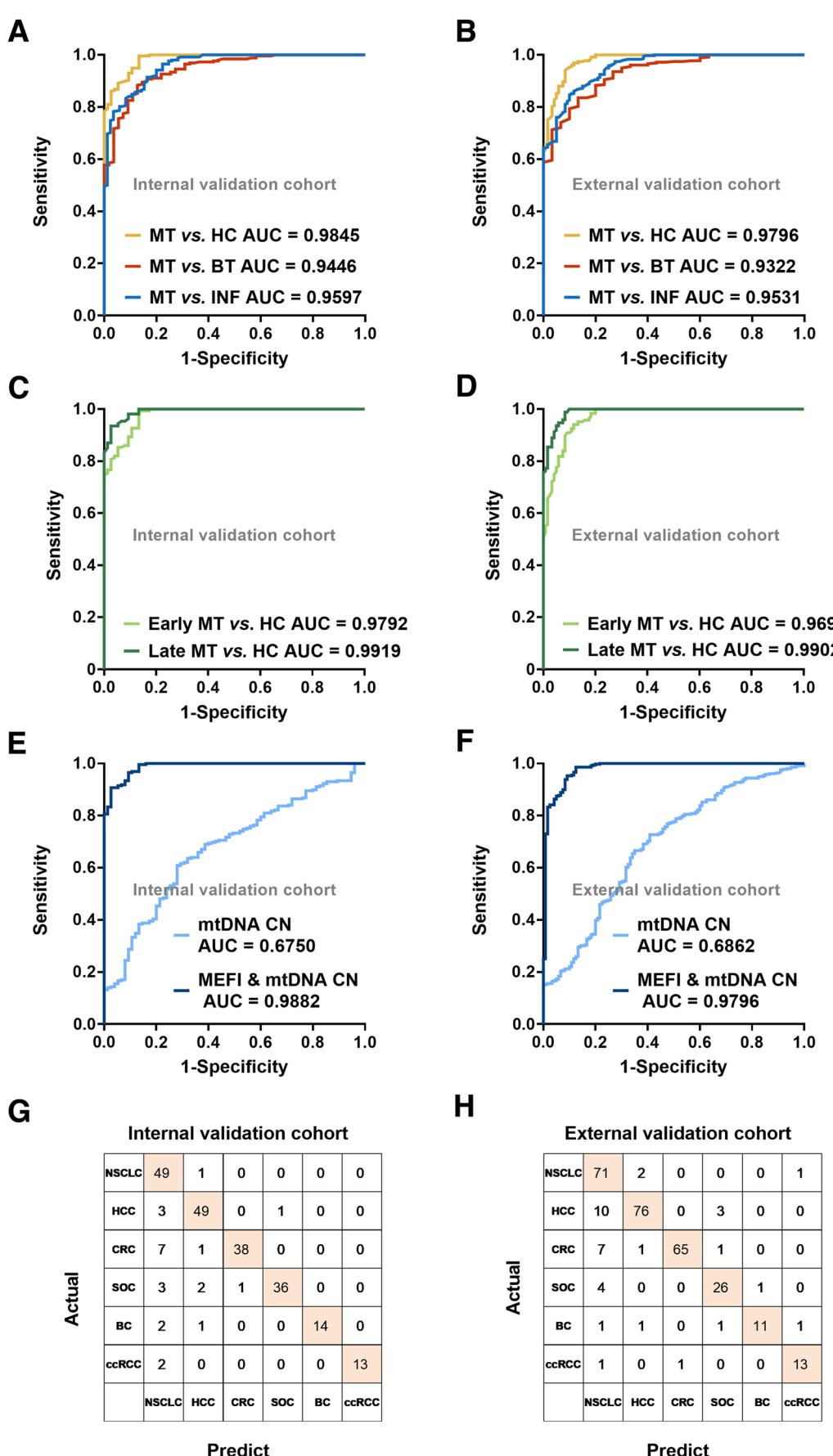

◄

**Figure 6.  Performance of cancer detection and tissue-of-origin classification models in the validation cohort.**

(A, B) Receiver operating characteristic curves (ROC) evaluating the overall performance of three cancer detection models in the internal validation cohort and external validation cohort. (C) ROC curves of MEFI score in the detection of MT at early ($n = 150$) and late stages ($n = 108$) in the internal validation cohort. (D) ROC curves of MEFI score in the detection of MT at early ($n = 187$) and late stages ($n = 172$) in the external validation cohort. (E, F) ROC curves of mtDNA copy number (CN) and MEFI score and mtDNA CN in the detection of MT when HC as a control in the internal validation cohort and external validation cohort. (G, H) Confusion matrix of the cancer patients for the tissue-of-origin prediction in the internal validation cohort and external validation cohort. MT malignant tumor, HC healthy control, BT benign tumor, INF inflammation, NSCLC non-small cell lung cancer, HCC hepatocellular carcinoma, CRC colorectal cancer, SOC serous ovarian cancer, BC breast cancer, ccRCC clear cell renal cell carcinoma, AUC area under the curve.

0.9277 (Appendix Fig. S16). Furthermore, the MEFI score exhibited high performance in differentiating both patients with MT at an early stage and those at a late stage from HC, with AUC of 0.9792, 0.9699, 0.9919, and 0.9902, respectively (Fig. 6C,D; Table EV3) in both the internal and external validation cohorts. Using leave-one-batch-out cross-validation, we also evaluated the performance of three models and found all AUC more than 0.9390 (Table EV4), which demonstrated the robustness of our models. As shown in Fig. 6E,F, the mtDNA CN had a lower performance (AUC = 0.6750 and 0.6862) than the MEFI score in distinguishing MT from HC in two validation cohorts. Furthermore, no significant difference in distinguishing MT from HC was found between the combination of the MEFI score and mtDNA CN and the MEFI score alone.

To explore whether the fragmentomic features of ccf-mtDNA were tumor type-specific, a cancer tissue-of-origin classification model was developed based on the MEFI score obtained using the random forest algorithm (Appendix Fig. S14). The confusion matrix showed that the MEFI score could predict the tissue-of-origin in most patients with cancer (Fig. 6G,H). As shown in Table EV5, our classification model had an accuracy of 89.24% (199 patients) for 223 patients in the internal validation cohort and 87.92% (262 patients) for 298 patients in the external validation cohort with different cancer types correctly detected at a specificity of 99.50%. The accuracy increased to 96.86% (216 patients) and 96.31% (287 patients) when classifying the tissue-of-origin according to the top two anatomic sites in two validation cohorts, respectively. And, the accuracy of the tissue-of-origin classification varied with tumor types. Taken together, our results demonstrate the principle of concept that the fragmentomic features of ccf-mtDNA can be used as effective biomarkers for the detection and tissue-of-origin classification of various cancers.

## Discussion

Herein, for the first time, we comprehensively characterized the fragmentomic features of ccf-mtDNA, explored the potential mechanisms underlying fragmentation, and observed aberrant fragmentomic features of ccf-mtDNA in patients with various types of cancer. Moreover, we demonstrated that the fragmentomic features of ccf-mtDNA represent a novel strategy for cancer detection and tissue-of-origin classification with high accuracy.

In this study, significantly different fragmentomic features of ccf-mtDNA with ccf-nDNA were characterized. In comparison with ccf-nDNA, ccf-mtDNA exhibited a shorter median fragment size with a larger coefficient of variation, presumably because the binding patterns of proteins bound to mitochondrial DNA were remarkably different from those bound to nuclear DNA (Lo et al,

2010). Considering that mtDNA is present in multiple cellular copies that may differ in number and sequence between individuals and tissues (Avital et al, 2012), it is reasonable to explain why the fragment size distribution of cf-mtDNA varies more among individuals. Additionally, ccf-DNA fragmentation is non-random, at least in part, and is mediated by various DNA nucleases that have a specific cutting preference (Han et al, 2020; Zhou et al, 2023). In the present study, we found that ccf-mtDNA exhibited stronger G- and A-end cleavage preferences than ccf-nDNA, which may be partially explained by the different intracellular locations, base compositions, and complex varieties of binding proteins between mtDNA and nDNA. MDS was calculated using 256 types of end motifs to represent the degree of randomization of the fragment cleavage patterns. The tRNA region with the largest FSD score had the lowest MDS, suggesting that more proteins may bind to the tRNA region. One unsolved question in this study is that the roles of many nucleases in ccf-mtDNA fragmentation remain elusive and could be explored in different nuclease knockout mice in the future. In addition, Ma et al also demonstrated the existence of circular mtDNA molecules in plasma (Ma et al, 2019), which cannot be analyzed using capture-based mtDNA sequencing. Furthermore, it is important to note that the mtDNA copy number of plasma samples detected in our study only represents the linear mtDNA.

Unlike the nuclear genome, mammalian mtDNA is coated mainly by mitochondrial transcription factor A (TFAM), which is an abundant protein present in ~1000 molecules per mtDNA molecule, or one TFAM protein molecule per 16 bp of mtDNA (Bonekamp and Larsson, 2018; Marom et al, 2019). Hence, the existence of an organized protein-DNA structure in the mitochondrial genome is plausible (Mishmar et al, 2019). Our results revealed a characteristic fragmentation profile with many obvious region-specific peaks across the mitochondrial genome, supporting the sequence preference for protein binding and indicating that the regions of the mitochondrial genome with lower protein binding are easily cleaved by endonucleases. Furthermore, ccf-nDNA molecules with different lengths are associated with different preferred DNA end sites. In our study, we observed larger 5′ end MDS in shorter fragments of ccf-mtDNA, suggesting the association of more enzyme cleavage with shorter fragments. Further investigation is required to elucidate the mechanisms underlying this association. Remarkably, the four functional regions (D-loop, mRNA, rRNA, and tRNA) of the mitochondrial genome exhibited different fragmentomic features of ccf-mtDNA, which may be caused by the different types, functions, and contents of proteins bound to the four functional regions. Furthermore, GC-rich nuclear chromatin has a more extended conformation than AT-rich chromatin (Dekker, 2007), suggesting that GC-rich regions may be more susceptible to enzyme digestion and produce shorter fragments. Our results also suggest that the base content of the mitochondrial

genome may affect ccf-mtDNA fragmentation. Furthermore, triple-stranded DNA may be the preferred substrate of the Endonuclease G, which is a nonspecific nuclease for all nucleic acid species and accounts for a large part of mitochondrial nuclease activity (Ohsato et al, 2002). Based on our results, Endonuclease G may play an important role in ccf-mtDNA fragmentation. Collectively, our results showed that the chromatin conformation of the mitochondrial DNA contributes to region-specific ccf-mtDNA fragmentation. Further studies are needed to determine the distinct types of ccf-mtDNA cleavage patterns and identify previously unknown mechanisms of ccf-mtDNA fragmentation.

Genome-wide fragmentation profiles of ccf-nDNA differ significantly between cancer and healthy controls (Cristiano et al, 2019). In our study, we first demonstrated the stability of the method for detecting ccf-mtDNA fragmentomic features by repeating the experiment three times. Subsequently, we found that storage time of peripheral blood for more than 12 h at 4 °C would lead to changes in the ccf-mtDNA fragmentation profile, which may be due to the lysis of leukocytes (Meddeb et al, 2019). In addition, temperature stress has been shown to activate blood platelets and cause mtDNA release (Dewitte et al, 2015). Therefore, proper plasma sample processing is of the utmost importance. Finally, aberrant ccf-mtDNA fragmentomic profiles were identified in patients with cancer. Thus far, the mechanisms underlying aberrant fragmentomic profiles remain unclear. Our results showed that patients with cancer exhibited significantly shorter fragments of ccf-mtDNA than non-cancer controls, and ccf-mtDNA fragments with tumor-specific mutations were significantly shorter than wild-type ccf-mtDNA fragments, suggesting that tumor-derived ccf-mtDNA fragments may at least partly account for the source of aberrant fragmentomic features. However, high levels of ccf-DNA observed in patients with cancer may not come from either neoplastic cells or from surrounding normal epithelial cells from the tumor's tissue-of-origin (Mattox et al, 2023), suggesting that cancer may have a systemic effect on protein-binding of the mitochondrial genome or plasma environments to influence ccf-mtDNA fragmentomics. Furthermore, we also observed the obvious changes of ccf-mtDNA fragmentomic features in the patients with INF/BT compared to those of healthy controls, which underlies the discriminatory ability of our detection model between non-malignant groups. Possible explanations are that certain diseases, such as hepatitis or liver cirrhosis, may exert a systemic influence on the chromatin conformation of the mitochondrial genome and the type or level of nucleases in plasma or tissues, which in turn leads to different fragmentomic features among non-malignant groups or between non-malignant and malignant groups. Moreover, the difference among patients with INF/BT and HC may contribute to a heightened specificity in cancer detection by reducing false positives. Our data also highlights the need for a nuanced interpretation of our results.

In the present study, we developed an easier-to-use approach (MEFI) for cancer detection and tissue-of-origin classification based on the fragmentomic features of ccf-mtDNA compared with DNA methylation-based methods, whose process remains complex. Similarly, a recent study developed a DELFI approach based on the fragmentation profile of ccf-nDNA (Mathios et al, 2021), which also showed good performance for cancer detection and tissue-of-origin classification. Unlike using low-depth WGS for detection, our cost-effective MEFI approach is based on capture-based

sequencing of ccf-mtDNA with high coverage, showing good repeatability for detecting fragmentomic features. Moreover, except for healthy individuals, patients with inflammatory diseases or benign tumors were enrolled in our study and used as non-cancer controls, representing a more important clinical setting for cancer detection. We also evaluated the accuracy of the MEFI approach in an external validation cohort, and the results underscore the robustness of our approach. Furthermore, the MEFI approach exhibited high performance in cancer detection in MT patients with early-stage cancer. Notably, we demonstrated the improved accuracy of the MEFI approach over existing approaches for the tissue-of-origin classification of six cancer types (Cohen et al, 2018; Mathios et al, 2021).

Although our results presented here are highly promising, we have to acknowledge two major limitations that need to be addressed in future studies. One is about the potential impact of uncharacterized confounding factors on the reported performance metrics. In addition, further evaluation of the MEFI approach is warranted in a larger cohort of patients with more types of cancer and even in a prospective cohort.

Overall, our study provides a ccf-mtDNA fragmentomics-based proof-of-principle strategy for cancer detection and tissue-of-origin classification. Our findings demonstrate that the fragmentomic features of ccf-mtDNA are sensitive and cost-effective biomarkers for the accurate non-invasive detection of cancer and, therefore, have the potential for important clinical translation.

# Methods

### Reagents and tools table

| Reagent/resource | Reference or source | Identifier or catalog number |
|---|---|---|
| **Oligonucleotides and other sequence-based reagents** | | |
| mtDNA capture probes | Zhou et al, 2020 | |
| PCR primers | Zhou et al, 2020 | |
| Human Cot-1 DNA™ | Thermo Fisher | Cat # 15279101 |
| **Chemicals, enzymes, and other reagents** | | |
| Dynabeads™ M-270 | Thermo Fisher | Cat # 65306 |
| Q5 High-Fidelity DNA Polymerase | NEB | Cat # M0491L |
| QIAquick MinElute kit | Qiagen | Cat # 28006 |
| dNTP | Takara | Cat # 4019 |
| **Software** | | |
| MedCalc software (v.19.4) | https://www.medcalc.org/ | |
| Fastp software (v.0.20.0) | https://github.com/OpenGene/fastp | |
| BWA software (v.0.7.1521) | https://github.com/lh3/bwa | |
| Picard Tools software (v.1.119) | https://broadinstitute.github.io/picard/ | |
| Picard Mark Duplicates software (v.1.81) | https://broadinstitute.github.io/picard/ | |

| Reagent/resource | Reference or source | Identifier or catalog number |
|---|---|---|
| Genome Analysis Toolkit 4 software (v.3.2-2) | https://gatk.broadinstitute.org/hc/en-us | |
| BEDTools software (v.2.26.0) | https://bedtools.readthedocs.io/en/stable/ | |
| Fastq-dump software (v.3.0.5) | https://www.ncbi.nlm.nih.gov/home/tools/ | |
| CellRanger-ATAC Count software (v.2.1.0) | https://www.10xgenomics.com/products/single-cell-atac | |
| Samtools software (v.1.15.1) | https://www.htslib.org/ | |
| Scipy software (v.1.6.2) | https://scipy.org/ | |
| R package "randomForest" (v.4.6-14) | https://cran.r-project.org/web/packages/randomForest/ | |
| GraphPad Software (v.9.5.1) | https://www.graphpad.com/ | |
| Other | | |
| QIAamp Circulating Nucleic Acid kit | Qiagen | Cat # 55114 |
| ENZA DNA kit | Omega | Cat # D3396 |
| NEB ultra v2 kit | NEB | Cat # E7645L |
| Qubit 3.0 | Thermo Fisher | |
| Illumina HiSeq XTen | Illumina | |

## Enrollment of patients and healthy individuals

Cancer patients with pathological diagnosis were enrolled. Patients with MT receiving antitumor treatment prior to enrollment and those with other types of concurrent tumors were excluded. The inclusion criteria for patients with BT or INF were as follows: (1) patients with BT or INF were diagnosed by pathological biopsy or patient's history, clinical manifestations and signs, laboratory test results, and imaging findings, respectively; (2) still cancer-free for at least 6 months of follow up. HCs were enrolled from individuals who underwent routine physical examinations based on inclusion criteria as follows: (1) with normal laboratory test results; (2) without a history of INF or BT within 5 years; (3) still cancer-free for at least 6 months of follow up. All participants were untreated at the time of enrollment, had complete clinical data and demographic data, and provided informed consent. The exclusion criteria for participants were as follows: (1) female participants who are pregnant or lactating or (2) with a history of cancer.

A total of 1607 participants were enrolled at Xijing Hospital (from March 2019 to February 2021) and Tangdu Hospital (from March 2019 to April 2021) of Fourth Military Medical University (FMMU), Xi'an. The disease cohort consisted of 877 patients with MT, 140 patients with BT, 290 patients with INF, and 300 healthy individuals. In detail, the MT cohort included 215 patients with NSCLC, 226 patients with HCC, 179 patients with CRC, 131 patients with SOC, 72 patients with BC, and 54 patients with ccRCC. The BT cohort included 40 patients with benign lung tumors (BLT), 42 patients with benign colonic neoplasms (BCN),

and 58 patients with benign ovarian tumors (BOT). The INF cohort included 70 patients with pneumonia (PN), 40 patients with inflammatory bowel disease (IBD), and 180 patients with hepatitis B (HB). The staging classification of NSCLC, CRC, BC, and ccRCC relies on the TNM staging system developed by the American Joint Committee on Cancer (AJCC), wherein early-stage includes stages I–II, and late-stage includes stages III–IV. The stratification of HCC follows the Barcelona Clinic Liver Cancer (BCLC) staging system, where early-stage HCC is defined as stages 0–A and late-stage HCC includes stages B–D. The staging classification of SOC is guided by the International Federation of Gynecology and Obstetrics (FIGO) guidelines, which classify early-stage as stages I–II and late-stage as stages III–IV. The demographic and clinicopathological characteristics of all the participants are listed in Tables EV6 and 7. The sample size was calculated using the MedCalc software (v.19.4) with 90% power, an alpha error of 5%, a beta error of 10%, and an estimated AUC of 0.9. Written informed consent was obtained from all participants, and the study was approved by the Ethics Committee of the Fourth Military Medical University (permission number: KY20183331-1; Date issued: 2018-03-08).

## Sample collection, DNA extraction, library construction, and sequencing

Sample collection was performed prior to treatment, and following the acquisition of diagnostic information. Whole blood samples (5 mL for each) were collected from patients with MT or BT 1 day before treatment and collected from patients with INF and HC at initial enrollment. To rule out the potential influence of sample processing on fragmentomic features analysis, plasma was separated by a standard protocol with two-step centrifugations as previously described (Zhou et al, 2022) within 2 h at 4 °C after collection and then immediately stored at −80 °C until the time of DNA extraction. Fresh tumor and paired para-tumor tissue samples were obtained from 40 patients with HCC at Xijing Hospital. The QIAamp Circulating Nucleic Acid kit (Qiagen, USA) and ENZA DNA kit (Omega, USA) were used for DNA extraction from plasma, peripheral blood mononuclear cells (PBMC), and fresh tissue samples, respectively, according to the manufacturer's manuals (Zhou et al, 2022). Qubit 3.0 (Thermo Fisher, USA) was used to determine the DNA quantity.

The WGS libraries were constructed for plasma cfDNA samples (10–20 ng for each sample) and genomic DNA samples (1 μg for each sample) from PBMC and fresh tissues of HCC patients by NEB ultra v2 kit (NEB, US) as previously described (Yin et al, 2019; Tao et al, 2020). The number of polymerase chain reaction (PCR) cycles during library construction was 12. To assess the repeatability and stability of the fragmentomic features identified using WGS, we randomly selected ten healthy individuals from 154 enrolled subjects from Xijing Hospital. For each plasma cfDNA sample from ten healthy individuals, we performed three times of library constructions and three times of WGS (Appendix Fig. S1a). In total, 50 WGS libraries from 30 HC (ten of them had 30 WGS libraries) and 90 WGS libraries from 30 HCC, 30 BOT, and 30 HB were sequenced on an Illumina HiSeq XTen platform using paired-end runs with 2 × 150 cycles (PE 150).

The mtDNA Cap-seq was performed using homemade biotinylated probes as previously described (Zhou et al, 2020; Liu et al, 2021) to analyze the linear mtDNA fragments in all plasma samples. Briefly,

ten plasma samples were pooled in a capture system, and a total of 2000 ng of WGS library from ten plasma samples (200 ng each sample) was mixed with 0.8 ng of mtDNA capture probes and hybridized at 65 °C for 24 h. In addition, the number of PCR cycles for mtDNA library amplification was set at 20. All the mtDNA libraries were sequenced on an Illumina HiSeq XTen platform using PE 150.

To minimize bias resulting from systematic errors in the experiment, we ensured that samples of different disease types are included in the same batch of experiments, and standardized experimental procedures were followed. The batching design of every sample included with clinical metadata at a sample level was shown in Dataset EV1. Additionally, all PCR reactions were conducted within the central region of the PCR thermal cycler's reaction module. Notably, all experiments were carried out by two individuals in a centralized laboratory, utilizing the same instruments across all samples.

## Sequencing data analysis

Illumina adapters and bases with quality scores below 30 were trimmed from the head and tail of each read in the raw sequencing data using the Fastp software (v.0.20.0) (Chen et al, 2018). To diminish contaminations from nuclear sequences of mitochondrial origin (NUMTs), we aligned qualified reads to both revised Cambridge reference sequence (rCRS) of the human mitochondrial DNA and human genome reference (hg19) using BWA software (v.0.7.1521) and reads only mapped to rCRS and with mapping quality (MAPQ) ≥20 were kept for further analyses. Picard Tools software (v.1.119) was used to sort the reads, and duplicate reads were removed using Picard Mark Duplicates software (v.1.81). Genome Analysis Toolkit 4 software (v.3.2-2) was used for local realignment. The final mtDNA coverage depths ranged from 39.84 X to 96.08 X for WGS data and ranged from 2354 X to 6814 X for capture-based mtDNA sequencing data (Tables EV8 and 9).

## MtDNA mutation calling

We first counted the read numbers of the major and minor alleles for each site of the mitochondrial genome and calculated the site-specific minor allele frequency (MAF). Then, the mtDNA mutation calling was carried out according to the following filter conditions: (1) at least three reads on each strand for the mutation site; (2) minimum MAF cutoff of 1%; (3) to remove heterogeneity sites in rCRS repeat regions (66–71, 303–311, 514–523, 12,418–12,425, 16,184–16,193); (4) to remove $C > A/G > T$ mutations with low MAF (<10%), which is known to arise from artificial guanine oxidation during sequencing library preparation; (5) to remove mtDNA mutations if the mutant rate and mutant base quality do not pass the binominal test ($P > 0.001$) (Campo et al, 2019). Tumor-derived mutations were defined as those present in fresh tumor tissue but not in paired leukocytes or fresh para-tumor tissue, whereas germline variants were defined as those present in both tumor tissue and paired control samples.

## Analysis of fragmentomic features

The fragment size of ccf-mtDNA was deduced from the coordinates of the aligned ends of the mitochondrial genome using the Picard tools software (v.1.119). Paired-end sequencing reads were first

recognized as fragments across the mitochondrial genome, and 5′ end base information was summarized into .bed files using the BEDTools software (v.2.26.0). Considering the different sequencing directions, different analytical methods were applied to the heavy and light strands of mtDNA. The end base for reads from the mtDNA light strand was considered the start of Read 1, whereas the end base for reads from the mtDNA heavy strand was defined as the complementary base of Read 1 termination. Then, the following formula was used to calculate the proportion of the 5′ end base: the number of 5′ specific-end fragments/sum of all 5′ A-end, 5′ G-end, 5′ C-end, and 5′ T-end fragments. The proportion of 5′ end 4-mer motifs was obtained in a similar manner. Additionally, the proportion of the 5′ end base was normalized based on the base composition of the mitochondria reference genome, and the 5′ end base preference was finally defined. To reflect the frequency distribution of 5′ end 4-mer motifs, we adopted the normalized Shannon entropy as a mathematical approach to calculate the 5′ end MDS using the following equation:

$$MDS = \sum_{i=1}^{256} -Pi^* \log(Pi)/\log(256)$$

where Pi is the frequency of the particular motif. The theoretical scale ranged from 0 to 1.

Next, we calculated the number of fragments covering each mtDNA site across the mitochondrial genome (1–16,569) and defined short and long fragments using the median fragment size of each sample. The FSD score at each mtDNA site was calculated using the coverage depth ratio of long to short fragments and corrected for GC content based on LOESS regression with a span setting of 0.75 and a degree setting of 1. The FSD score was scaled into a z-score by comparing the variable values with the overall mean value. Finally, the fragmentation profile at 1–16,569 sites for each ccf-mtDNA sample was depicted based on the FSD score. When analyzing the fragmentomic features of the D-loop, mRNA, rRNA, and tRNA regions, the fragments only with 5′ end falling within the given region were counted. In addition, the influence of whole blood sample placement time at 4 °C on the ccf-mtDNA fragmentation profile was evaluated based on the mtDNA Cap-seq data of 5 HCs generated in our previous study (Liu et al, 2021).

## Analysis of public mtscATAC-seq data

The mitochondrial single-cell ATAC-seq (mtscATAC-seq) data were obtained from GSE142745 (Lareau et al, 2021). Firstly, the Fastq-dump software (v.3.0.5) was employed for converting sequence read archive (SRA) files into FASTQ format. And, the raw sequencing reads were aligned to the hg19 reference genome using CellRanger-ATAC Count software (v.2.1.0). Based on the .bam file generated in the previous step, we calculated the mtDNA depth using Samtools software (v.1.15.1), which was also scaled into the z-score by comparing the variable value against the overall mean value. Finally, the standardized mtDNA depth was obtained from 30 randomly selected single cells.

## Calculation of mtDNA copy number

Based on our established method, the average sequencing depth of mtDNA and the average sequencing depth of the nuclear reference

gene were used to calculate the relative mtDNA copy number (Zhou et al, 2020). And the following formula was used to calculate the mtDNA copy: 2* average depth of mtDNA/average depth of nuclear reference gene.

## Extraction of fragmentomic features and construction of machine learning models

Three categories of ccf-mtDNA fragmentomic features, including fragmentation profile, fragment size, and 5′ end motif were extracted from the mtDNA Cap-seq data as follows. First, Spearman's rank correlation coefficient between the ccf-mtDNA fragmentation profile of each sample and the median profile of the aforementioned 30 HC plasma samples was calculated. Then, the peak on the fragmentation profile was defined by setting the peak width ≥5 bp. Compared with the peaks in the median HC profile, new peaks were defined as those with a distance of more than 20 bp. The number of new peaks was then determined. The mitochondrial genome was equally divided into 255 windows. The areas of the positive and negative peaks were calculated for each window at a baseline FSD score of 0. Then, the Euclidean distance from each fragmentation profile to the median HC profile was calculated within each window using the Scipy software (v.1.6.2). The median fragment size of each ccf-mtDNA sample was then determined. Finally, except for 5′ end base preference, 256 different types of 5′ end 4-mer motifs were identified, and their percentages were calculated. Based on 5′ end motifs, the 5′ end MDS was also calculated.

To construct machine learning models for cancer detection and the origin of tissue classification, a total of 948 participants from Xijing Hospital were randomly divided into training and internal validation cohorts in a 1:1 ratio. The training cohort consisted of 475 individuals, including 260 patients with MT (60 NSCLC, 63 HCC, 46 CRC, 49 SOC, 25 BC, and 17 ccRCC), 55 patients with BT (15 BLT, 15 BCN, and 25 BOT), 85 patients with INF (20 PN, 15 IBD, and 50 HB), and 75 HCs. The internal validation cohort consisted of 473 individuals, including 258 patients with MT (60 NSCLC, 63 HCC, 46 CRC, 48 SOC, 25 BC, and 16 ccRCC), 55 patients with BT (15 BLT, 15 BCN, and 25 BOT), 85 patients with INF (20 PN, 15 IBD, and 50 HB), and 75 HCs. A total of 359 patients with MT (95 NSCLC, 100 HCC, 87 CRC, 34 SOC, 22 BC, and 21 ccRCC), 30 patients with BT (10 BLT, 12 BCN, and 8 BOT), 120 patients with INF (30 PN, 10 IBD, and 80 HB), and 120 HCs from Tangdu Hospital were used as the external validation cohort.

The training cohort was used to build cancer detection models based on all fragmentomic features, using the random forest algorithm in the R package "randomForest" (v.4.6-14). The number of trees was set to 500, and the number of features was set to six for each tree. Three cancer detection models were constructed using HC, INF, and BT as controls, respectively. A tenfold cross-validation was performed to evaluate the models' performance in the training cohort. Two validation cohorts were used to evaluate the performance of all models. Furthermore, leave-one-batch-out, cross-validation was also used to evaluate the performance of all models.

The individuals employed for the development and validation of the tissue-of-origin classification model consisted of the patients who were accurately classified as MT with 99.5% specificity using the cancer detection model with HC as the control group, including 239 MT in the training cohort, 223 MT in the internal validation cohort, and 298 MT in the external validation cohort. Subsequently, a model was trained to classify the tissue-of-origin using the

### The paper explained

**Problem**

Cancer is a major cause of death worldwide. Currently, diagnostics and treatment options for cancer mainly rely on imaging, pathology, and serology, and are thus highly limited. Remarkable advancements have been achieved in the realm of liquid biopsy for cancer diagnosis, stemming from the development of groundbreaking blood-based cell-free DNA (cfDNA) biomarkers. However, there is an unmet need for novel biomarkers that allow non-invasive detection of multi-cancer and tissue-of-origin classification.

**Results**

In our study, we found that ccf-mtDNA exhibits different fragmentomic features, including fragment size, 5′ end base preference, and 5′ end motif diversity from ccf-nDNA. We also observed region-specific fragmentomic features of ccf-mtDNA, which was associated with protein binding, base composition, and special structure of mitochondrial DNA. Furthermore, based on ccf-mtDNA fragmentomic features, we developed cancer detection models and tissue-of-origin classification model, which exhibited good performance in two validation cohorts.

**Impact**

Our high-performing and low-cost approach based on aberrant ccf-mtDNA fragmentomic features could enable clinicians to identify cancer patients in their early stages with unprecedented sensitivity and specificity, leading to earlier interventions and improved treatment outcomes.

random forest algorithm in the R package "randomForest" (v.4.6-14), based on all fragmentomic features and with the same parameters as the cancer detection models. Two validation cohorts were used to evaluate the performance of the tissue-of-origin classification model.

## Statistical analysis

GraphPad Prism v.9.5.1 (GraphPad Software) was used for all statistical analyses. Student's *t*-test, Wilcoxon's rank test, and Friedman test were used to compare the difference between two or among more groups with continuous variables. The correlations between the two groups were measured using Spearman's rank correlation coefficient. All *P* values were two-tailed and reported using a significance level of 0.05. ROC curves were constructed to evaluate the diagnostic capacity of three cancer detection models. The cutoff value was determined according to Youden's index. Based on true-positive (TP), true-negative (TN), false-positive (FP), and false-negative (FN) of cancer prediction, the sensitivity [TP/(TP + FN)] and specificity [TN/(TN + FP)] as well as the corresponding 95% confidence intervals were calculated. To visualize the accuracy of the tissue-of-origin classification model, confusion matrix was built by sklearn package "confusion matrix" (v.0.22.1).

## Data availability

The raw sequencing data underlying this study is available at the BIG Data Center, Beijing Institute of Genomics (BIG), with accession number PRJCA020284 (https://ngdc.cncb.ac.cn/bioproject/browse/PRJCA020284). All codes used for the analyses and visualizations in

this manuscript are available at https://github.com/Mitoomics/MEFI_code.

The source data of this paper are collected in the following database record: biostudies:S-SCDT-10_1038-S44321-024-00163-6.

## Peer review information

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

## Acknowledgements

This work was supported by the National Natural Science Foundation of China (grant no. 82330073 and 82402731), the Science & Technology Coordination and Innovation Project of Shaanxi Province, China (grant no. 2023-ZDLSF-46), and the Clinical Research Program of Air Force Medical University (grant no. 2021LC2104).

## Author contributions

**Yang Liu**: Conceptualization; Data curation; Software; Formal analysis; Validation; Investigation; Visualization; Methodology; Writing—original draft; Project administration; Writing—review and editing. **Fan Peng**: Conceptualization; Data curation; Software; Formal analysis; Validation; Investigation; Visualization; Methodology; Writing—original draft; Project administration. **Siyuan Wang**: Conceptualization; Data curation; Software; Validation; Investigation; Visualization; Methodology; Writing—original draft. **Huanmin Jiao**: Data curation; Software; Formal analysis; Investigation; Visualization; Methodology. **Miao Dang**: Data curation; Formal analysis; Investigation. **Kaixiang Zhou**: Data curation; Formal analysis; Supervision; Validation. **Wenjie Guo**: Data curation; Supervision; Validation. **Shanshan Guo**: Data curation; Software; Formal analysis; Supervision; Investigation. **Huanqin Zhang**: Data curation; Supervision; Investigation; Project administration. **Wenjie Song**: Data curation; Supervision; Validation; Visualization; Project administration. **Jinliang Xing**: Conceptualization; Resources; Data curation; Software; Formal analysis; Supervision; Funding acquisition; Validation; Investigation; Visualization; Methodology; Writing—original draft; Project administration; Writing—review and editing.

Source data underlying figure panels in this paper may have individual authorship assigned. Where available, figure panel/source data authorship is listed in the following database record: biostudies:S-SCDT-10_1038-S44321-024-00163-6.

## Disclosure and competing interests statement

The authors declare no competing interests.

