## [Peer Review File · EMBO Molecular Medicine]

Aberrant fragmentomic features of circulating cell-free mitochondrial DNA as novel biomarkers for multi-cancer detection

Yang Liu, Fan Peng, Siyuan Wang, Huanmin Jiao, Miao Dang, Kaixiang Zhou, Wenjie Guo, Shanshan Guo, Huanqin Zhang, Wenjie Song, and Jinliang Xing

Corresponding author: Jinliang Xing (xingjl@fmmu.edu.cn)

Review Timeline:

Submission Date:	30th May 24
Editorial Decision:	25th Jun 24
Revision Received:	22nd Aug 24
Editorial Decision:	16th Sep 24
Revision Received:	27th Sep 24
Accepted:	18th Oct 24

Editor: Jingyi Hou

Transaction Report:

25th Jun 2024

Dear Prof. Xing,

Thank you again for submitting your work to EMBO Molecular Medicine. We have now heard back from the three referees who evaluated your manuscript. As you will see below the referees raise substantial concerns, which preclude the publication of the manuscript in its current form.

Referees #2 and #3 think without key information about the study design, the validity of the conclusion cannot be properly assessed. Nevertheless, the referees appreciate the potential interest of the findings and have provided quite constructive and detailed suggestions. Taken together and considering that the referees acknowledge the overall relevance, we would like to offer you a chance to address these concerns in a major revision. Please note that the revised manuscript will be reevaluated and we cannot accept it for publication if the raised concerns, especially those regarding the experimental design, are not satisfactorily addressed.

We would welcome the submission of a revised version within three months for further consideration. As you may already know, our editorial policy allows in principle a single round of major revision, and it is therefore essential to provide responses to the reviewers' comments that are as complete as possible.

I look forward to receiving your revised manuscript.

Kind regards,
Jingyi

Jingyi Hou
Editor
EMBO Molecular Medicine

We require:

- 1) A .docx formatted version of the manuscript text (including legends for main figures, EV figures and tables). Please make sure that the changes are highlighted to be clearly visible.
- 2) Individual production quality figure files as .eps, .tif, .jpg (one file per figure). For guidance, download the 'Figure Guide PDF': (<https://www.embopress.org/page/journal/17574684/authorguide#figureformat>).
- 3) A .docx formatted letter INCLUDING the reviewers' reports and your detailed point-by-point responses to their comments. As part of the EMBO Press transparent editorial process, the point-by-point response is part of the Review Process File (RPF), which will be published alongside your paper.
- 4) A complete author checklist, which you can download from our author guidelines

(<https://www.embopress.org/page/journal/17574684/authorguide#submissionofrevisions>). Please insert information in the checklist that is also reflected in the manuscript. The completed author checklist will also be part of the RPF.

6) It is mandatory to include a 'Data Availability' section after the Materials and Methods. Before submitting your revision, primary datasets produced in this study need to be deposited in an appropriate public database, and the accession numbers and database listed under 'Data Availability'. Please remember to provide a reviewer password if the datasets are not yet public (see <https://www.embopress.org/page/journal/17574684/authorguide#dataavailability>).

12) For more information: There is space at the end of each article to list relevant web links for further consultation by our readers. Could you identify some relevant ones and provide such information as well? Some examples are patient associations,

relevant databases, OMIM/proteins/genes links, author's websites, etc...

13) Author contributions: You will be asked to provide CRediT (Contributor Role Taxonomy) terms in the submission system. These replace a narrative author contribution section in the manuscript.

14) A Conflict of Interest statement should be provided in the main text.

Please also suggest a striking image or visual abstract to illustrate your article as a PNG file 550 px wide x 300-800 px high.

***** Reviewer's comments *****

Referee #1 (Comments on Novelty/Model System for Author):

The study achieves high marks in technical quality and novelty by thoroughly characterizing ccf-mtDNA and revealing its significant association with cancer through comprehensive fragmentomic analysis. However, the medical impact is somewhat constrained by the limited number of patients, especially in common cancers like breast cancer. Expanding the sample size would enhance the study's ability to generalize findings and improve diagnostic applicability.

Referee #1 (Remarks for Author):

The paper is well-written and provides an engaging study on the use of mtccfDNA for cancer detection, a valuable addition to the field of liquid biopsy. The researchers generated sequencing data from 1,323 plasma samples and found that fragmentomic features of ccf-mtDNA differ significantly from those of circulating cell-free nuclear DNA. The study developed cancer detection models based on these features, achieving high accuracy and good capacity in distinguishing cancer patients from non-cancer controls.

The paper's findings are novel and compelling, and I am eager to accept the paper with minor changes.

Minor points:

- In the introduction, the authors should describe other technologies used to detect ctDNA that do not require bisulfite modification, such as MeDIP.
- In the main Figure 1, the number of samples tested must be specified.
- The observation that the fragmentation patterns of the ccf-mtDNA are modified depending on storage timing at 4 degrees is an important observation that is not further discussed. This has significant implications for potential clinical implementation. Any thoughts or further discussion on this aspect is needed.
- The acronyms "MT," "BT," and "INF" should be defined when first mentioned. Please check all acronyms in graphs and text, as some definitions are missing.
- For the tissue-of-origin classification model, clarification is needed regarding the inclusion criteria for patients: were early-stage or late-stage cancers included? Additionally, why did the authors reduce the number of breast cancer patients from an initial 40 to 22 when combining internal and external validation cohorts? It would be beneficial to include a graphical schema/table illustrating which samples were used for different analyses to enhance understanding.

Referee #2 (Remarks for Author):

The authors present an interesting manuscript focused on ccf-mtDNA (cell-free mitochondrial DNA) a distinct subset of cell-free

DNA which they attempt to show has fragmentomic differences in cancer, distinct from those found in nuclear-derived cfDNA. The findings are interesting and could potentially have impact for cancer detection and tissue of origin determination; however, my enthusiasm is significantly limited by several technical concerns that make it impossible to determine whether the demonstrated performance is artifactual or true biological signal.

Major Comments:

- The fragmentomic analysis of mitochondrial cfDNA is insufficiently described. It is not clear how the 30 controls with "high reproducibility in detecting ccf-mtDNA fragmentomics" were selected from the full 154 controls, and what this means. Why was the coverage so different between ccf-mtDNA and ccf-nDNA? If this subset was selected as a limited set for initial feature exploration it would make more sense to perform the same high coverage sequencing to avoid introduction of artifacts.
- The authors describe GC content differences impacting mitochondrial fragments, but then do not seem to employ a GC correction in computing fragmentomic metrics for the classifier. This is a known bias introduced by DNA library construction, and by the authors own admission is especially relevant in mtDNA. GC correction is standard in many of the fragmentomic approaches the authors cite (Cristiano et. al., Mathios et. al. etc), why is it not employed here?
- Insufficient information is provided on pre-analytical and technical sources of variation. What batching scheme was used in library preparation? The authors note that time to processing impacts fragmentation profile so all samples were processed within 2 hours, but no evidence is provided to suggest that other variables (library batch, sample site, PCR cycles) were adequately controlled for in the study design. This makes it impossible to conclude that the classifier performance observed is due to biological and not technical differences. In addition to more detailed description in the methods, I think it would be helpful for the authors to provide clear supplementary tables indicating the sample site and batching design of every sample included with clinical metadata at a sample level rather than only in aggregate.
- Were cross-validation and validation cohorts processed in a temporally separate way, or sequenced together? The classifier performance the authors report is indeed very impressive, but without more technical details it is difficult to evaluate.
- Why did the authors train separate classifiers for the HC, BT and INF non-cancer groups? Given the difficulties in identifying benign lesions in the general population, separating this group out seems unrealistic. Separating out inflammatory diseases could be a reasonable strategy, but the question of why cfDNA profiles are so different in this group as compared to healthy controls if the differences the authors describe are meant to be tumor-specific is not sufficiently explained in the manuscript.

Minor Comments:

- some grammatical errors throughout - Not the priority of revision and did not significantly impede my review of the manuscript, but the authors may wish to copy edit more thoroughly prior to publication if accepted.
- would be good to provide more details on processing pipeline for ATAC-seq data

Referee #3 (Comments on Novelty/Model System for Author):

It's interesting to evaluate if deeper sequencing of mtDNA in cfDNA could be useful for cancer detection and tissue of origin, however I don't think this manuscript can be properly assessed without information on experimental design, such as batching scheme. Otherwise, it seems as though the performance of cancer detection could just be a batch effect.

Referee #3 (Remarks for Author):

"Furthermore, region-specific fragmentomic features of ccf- mtDNA were observed, which was associated with protein binding, base composition and special structure of mitochondrial DNA."

Using the public ATAC seq data to confirm this hypothesis, and there are some regions as shown in figure 2B that indicate this negative correlation, but on the whole, I don't see much relationship. Consider other approaches for visualizing the data to support this conclusion, such as a scatter plot of ATAC seq depth vs FSD score.

While it's true that there is a modest difference in fragment length distribution by the four functional groups, it's not clear if the difference is in the expected direction. Further, it's not clear if this difference in fragmentation is biologically relevant.

"When comparing to non-cancer controls, six types of cancer patients exhibited aberrant fragmentomic features."

You claim that the analysis shows similar fragmentation profiles between HC, BT, and INF groups, however in 5B show that the distribution of correlation to median NC is significantly different between HC and BT and HC and INF. Suggest re-wording.

Please clarify how mtDNA copy number is calculated.

"Both internal and external validation cohorts demonstrated the excellent capacity of our model in distinguishing cancer patients from non-cancer control, with all area under curve higher than 0.9110. The overall accuracy of tissue-of-origin was 90.59% and 90.45% for six cancer types in two validation cohort, respectively."

Consider showing showing scores by stage and tissue type in a figure, as well as breaking out tables by stage I, II, etc.

"We also evaluated the accuracy of MEFI approach in an independent external validation cohort" it seems a stretch to call this an independent external validation cohort. Patients enrolled at the same time, in the same country, and presumably were processed with the assay at the same time. Please clarify why this is an independent external validation cohort.

Please clarify the source of the healthy individuals. You mention that all donors enrolled from the 2 different hospitals: how did healthy individuals enroll through a hospital? Were these healthy staff volunteers, or some other source?

Please explain how samples are allocated to extraction, library, and sequencing batch. It's unclear if confounding by batching may be responsible for the high performance seen in cancer detection and/or tissue of origin.

Please clarify the experimental design of the technical replicates and provide further information on results. I was not able to follow the conclusion.

I looked at the linked GitHub repository, but repository defaulted to main which has no code. The master branch has the code instead. Please correct the branching so that it's obvious where to find the code. Further, the code for the random forest classifier, just shows fitting a random forest to a dataset, but does not demonstrate the described cross-validation. Please update to include the cross-validation. It would also be helpful to have the already processed features in a CSV in order to verify the findings.

Multiple times, the authors say that Mathios 2021 performed tissue of origin classification, but this is inaccurate. Please correct.

Please correct spelling of wild-type in figure 5F and 5G.

August 21, 2024

Dr. Jingyi Hou

Editor-in-Chief

EMBO Molecular Medicine

Dear Editor,

We greatly thank the editor and reviewers for the constructive comments on our manuscript entitled by “Aberrant fragmentomic features of circulating cell-free mitochondrial DNA as novel biomarkers for multi-cancer detection” (EMM-2024-20082). We have carefully addressed all concerns. The point-by-point responses to the comments are provided below. We hope that our response and revision can fulfill the requirement for publication.

If there are any further questions, please feel free to contact me by email at xingjl@fmmu.edu.cn, or by phone at (86) 29-84774551, or by FAX (86) 29-84774551.

Sincerely,

Jinliang Xing, M.D., Ph.D.

State Key Laboratory of Holistic Integrative Management of Gastrointestinal Cancers and Department of Physiology and Pathophysiology, Fourth Military Medical University, Xi'an, Shaanxi 710032, P. R. China.

Claim: All changes made in the revised manuscript and supplementary materials are highlighted in yellow.

Point-by-point response to reviewers

Referee #1 (Comments on Novelty/Model System for Author):

The study achieves high marks in technical quality and novelty by thoroughly characterizing ccf-mtDNA and revealing its significant association with cancer through comprehensive fragmentomic analysis. However, the medical impact is somewhat constrained by the limited number of patients, especially in common cancers like breast cancer. Expanding the sample size would enhance the study's ability to generalize findings and improve diagnostic applicability.

Response: We totally agree with the reviewer's comments. As a proof-of-concept study, the present study just only validates the potential application of ccf-mtDNA fragmentomic features in pan-cancer detection and tissue-of-origin classification. Therefore, six cancer types were enrolled, with not many samples for each cancer type. The establishment of detection model for specific cancer types is undergoing based on larger cancer cohort. Here, based on reviewer's suggestion, we have expanded the sample size (from 1,323 to 1,607). The detailed information of 284 new samples was shown in following Table. The experimental process, sequencing platform and data analysis pipeline of the new samples were

kept to be consistent with those of the previous samples. And, we reanalyzed the data of all samples. The updated data indicated almost same results.

Disease type	Xijing Hospital	Tangdu Hospital
	01/2021-02/2021 (n)	03/2021-04/2021 (n)
Malignant tumor	118	94
Non-small cell lung cancer	20	15
Hepatocellular carcinoma	26	20
Colorectal cancer	32	27
Serous ovarian cancer	17	14
Breast cancer	10	7
Clear cell renal cell carcinoma	13	11
Benign tumor	30	0
Benign lung tumor	16	0
Benign colonic neoplasm	14	0
Benign ovarian tumor	0	0
Inflammatory diseases	16	0
Pneumonia	0	0
Inflammatory bowel disease	16	0
Hepatitis B	0	0
Healthy control	26	0
Total	190	94

Referee #1 (Remarks for Author):

The paper is well-written and provides an engaging study on the use of mtccfDNA for cancer detection, a valuable addition to the field of liquid biopsy. The researchers generated sequencing data from 1,323 plasma samples and found that fragmentomic features of ccf-mtDNA differ significantly from those of circulating cell-free nuclear DNA. The study developed cancer detection models based on these features, achieving

high accuracy and good capacity in distinguishing cancer patients from non-cancer controls.

The paper's findings are novel and compelling, and I am eager to accept the paper with minor changes.

Minor points:

- In the introduction, the authors should describe other technologies used to detect ctDNA that do not require bisulfite modification, such as MeDIP.

Response: Based on reviewer's suggestion, we have added related description on additional ctDNA methylation detection technique in Introduction section on Page 3 Lines 41-43.

- In the main Figure 1, the number of samples tested must be specified.

Response: The information about the number of samples tested has been added in revised Figure 1.

- The observation that the fragmentation patterns of the ccf-mtDNA are modified depending on storage timing at 4 degrees is an important observation that is not further discussed. This has significant implications for potential clinical implementation. Any thoughts or further discussion on this aspect is needed.

Response: We have added further discussion on this aspect in revised manuscript on Page 19-20 Lines 388-395 as following: In our study, we found that storage time of peripheral blood for more than 12 h

at 4 °C would lead to changes in ccf-mtDNA fragmentation profile, which may be due to the lysis of leukocytes (Meddeb *et al*, 2019). In addition, temperature stress has been shown to activate blood platelets and cause mtDNA release (Dewitte *et al*, 2015). Therefore, proper plasma sample processing is of the utmost importance."

- The acronyms "MT," "BT," and "INF" should be defined when first mentioned. Please check all acronyms in graphs and text, as some definitions are missing.

Response: We have made related revision in the revised manuscript (highlighted in yellow).

- For the tissue-of-origin classification model, clarification is needed regarding the inclusion criteria for patients: were early-stage or late-stage cancers included? Additionally, why did the authors reduce the number of breast cancer patients from an initial 40 to 22 when combining internal and external validation cohorts? It would be beneficial to include a graphical schema/table illustrating which samples were used for different analyses to enhance understanding.

Response: Given that the model is learned on a relatively pure data set, in practical applications, tissue-of-origin classification model is usually required to classify patients who have already been diagnosed with tumors. Therefore, it is more aligned with real-world application scenarios using the data of the cancer patients who were identified as MT

by the cancer detection model to construct tissue-of-origin classification model. As previously described (Cristiano *et al*, 2019; Bao *et al*, 2022), with HC as the control group, the cancer patients who were identified as malignant tumors (MT) with 99.5% specificity by the cancer detection model were further used to construct a more accurate tissue-of-origin classification model. Therefore, the number of patients with MT was inconsistent between cancer detection model and tissue-of-origin classification model. Detailed information was shown in following Table. To make this point more clearly, we also have added more detailed description on construction and validation of the tissue-of-origin classification model in Methods section on Page 32 Lines 659-664 and added a graphical schema in Appendix Figure S14.

	Cancer type	Cancer detection model (n)	Tissue-of-origin classification model (n)
Training cohort	NSCLC	60	54
	HCC	63	55
	CRC	46	46
	SOC	49	45
	BC	25	24
	RCC	17	15
	Total	260	239
Internal validation cohort	NSCLC	60	50
	HCC	63	53
	CRC	46	46
	SOC	48	42
	BC	25	17
	RCC	16	15
	Total	258	223

	NSCLC	95	74
	HCC	100	89
	CRC	87	74
External validation cohort	SOC	34	31
	BC	22	15
	RCC	21	15
	Total	359	298

Abbreviations: NSCLC, non-small cell lung cancer; HCC, hepatocellular carcinoma; CRC, colorectal cancer; SOC, serous ovarian cancer; BC, breast cancer; ccRCC, clear cell renal cell carcinoma.

Referee #2 (Remarks for Author):

The authors present an interesting manuscript focused on ccf-mtDNA (cell-free mitochondrial DNA) a distinct subset of cell-free DNA which they attempt to show has fragmentomic differences in cancer, distinct from those found in nuclear-derived cfDNA. The findings are interesting and could potentially have impact for cancer detection and tissue of origin determination; however, my enthusiasm is significantly limited by several technical concerns that make it impossible to determine whether the demonstrated performance is artifactual or true biological signal.

Major Comments:

-The fragmentomic analysis of mitochondrial cfDNA is insufficiently described. It is not clear how the 30 controls with "high reproducibility in detecting ccf-mtDNA fragmentomics" were selected from the full 154 controls, and what this means. Why was the coverage so different between ccf-mtDNA and ccf-nDNA? If this subset was selected as a

limited set for initial feature exploration it would make more sense to perform the same high coverage sequencing to avoid introduction of artifacts.

Response: To evaluate the repeatability of our detecting procedure for plasma ccf-mtDNA fragmentomic features, 10 healthy individuals (HCs) were randomly selected and cfDNA sample was subject to three times of library construction and WGS (Appendix Figure S1a). No significant difference was observed in the detection of ccf-mtDNA fragmentomics features among the three replicates, which also suggested no notable bias among different test batches (all $P > 0.05$, Appendix Figure S1b). Then, WGS data were obtained from 30 randomly selected plasma samples from healthy individuals for comparing the fragmentomic features of ccf-mtDNA (average sequencing coverage = 71.91 X) and ccf-nDNA (average sequencing coverage = 8.93 X). To make this point more clearly, we have made related revision in Result section (Page 6 Line 92-104). Considering two copies of nuclear DNA (nDNA) and 10^3 to 10^4 copies of mtDNA in each cell, it is reasonable to observe so different coverage between ccf-mtDNA and ccf-nDNA in WGS data of plasma samples. With $3 * 10^9$ bp of nuclear genome and 16569 bp length of mitochondrial genome in length, nDNA data will have much more sequencing reads than mtDNA data at same high sequencing coverage, ultimately leading to the great bias of fragmentomic feature analysis. In

addition, our data indicated the stable fragmentomic features when ccf-nDNA and ccf-mtDNA data were over 1 X and 50 X, respectively (Appendix Figure S2). So, we did not replace the original data in revised manuscript.

-The authors describe GC content differences impacting mitochondrial fragments, but then do not seem to employ a GC correction in computing fragmentomic metrics for the classifier. This is a known bias introduced by DNA library construction, and by the authors own admission is especially relevant in mtDNA. GC correction is standard in many of the fragmentomic approaches the authors cite (Cristiano et. al., Mathios et. al. etc), why is it not employed here?

Response: Sorry for our unclear description and causing such a misunderstanding. In fact, GC correction was employed when the fragmentation profile was analyzed in this study. To make it more clear, we have made related revision in Methods section on Page 29 Lines 585-589.

-Insufficient information is provided on pre-analytical and technical sources of variation. What batching scheme was used in library preparation? The authors note that time to processing impacts fragmentation profile so all samples were processed within 2 hours, but no evidence is provided to suggest that other variables (library batch, sample site, PCR cycles) were adequately controlled for in the study

design. This makes it impossible to conclude that the classifier performance observed is due to biological and not technical differences. In addition to more detailed description in the methods, I think it would be helpful for the authors to provide clear supplementary tables indicating the sample site and batching design of every sample included with clinical metadata at a sample level rather than only in aggregate.

Response: As reviewer mentioned, pre-analytical and technical sources of variation should be carefully considered in the fragmentomic analysis of ccf-mtDNA. In our study design, we have indeed tried our best to adequately control the variables which impact fragmentation profile, such as time to processing of samples, library batch, sample site, and PCR cycles. Therefore, to minimize potential bias resulting from systematic errors, samples of different disease types were included in the same batch of experiments, and standardized experimental procedures were followed. Additionally, all PCR reactions were conducted within the central region of the PCR thermal cycler's reaction module. Notably, the experiments were performed by utilizing the same instruments across all samples in a centralized laboratory. No obvious batch effect was found. Based on the reviewer's suggestion, we have added more detailed description in Methods section on Page 25 Lines 502-511, 515-519 and Page 26 Lines 522-530. We also provided Table EV3 indicating the sample site and batching design of every sample.

- Were cross-validation and validation cohorts processed in a temporally separate way, or sequenced together? The classifier performance the authors report is indeed very impressive, but without more technical details it is difficult to evaluate.

Response: Based on the reviewer's suggestion, we have provided more technical details in Method section on Page 25-26 Lines 522-530 as following: To minimize bias resulting from systematic errors in the experiment, we ensured that samples of different disease types are included in the same batch of experiments, and standardized experimental procedures were followed. Additionally, all PCR reactions were conducted within the central region of the PCR thermal cycler's reaction module. Notably, all experiments were carried out by two individuals in a centralized laboratory, utilizing the same instruments across all samples. The batching design of every sample included with clinical metadata at a sample level was shown in revised Table EV3.

-Why did the authors train separate classifiers for the HC, BT and INF non-cancer groups? Given the difficulties in identifying benign lesions in the general population, separating this group out seems unrealistic. Separating out inflammatory diseases could be a reasonable strategy, but the question of why cfDNA profiles are so different in this group as compared to healthy controls if the differences the authors describe are meant to be tumor-specific is not sufficiently explained in the manuscript.

Response: Thank for the reviewer's comment. As a proof of concept, the present study emphasizes the potential application of ccf-mtDNA fragmentomic features in cancer detection and tissue-of-origin classification. Early or differential diagnosis of cancer is of great importance in clinical oncology, depending the various cancer types. For example, the differential diagnosis of benign and malignant lung diseases is a critical challenge in clinical practice. The early diagnosis of HCC is great importance for overall survival in patients with background diseases of hepatitis or liver cirrhosis. Therefore, our principle-of-concept study established separate classifiers for the HC, BT and INF non-cancer groups in training cohort and evaluated the performance. The establishment of detection model for specific cancer types is undergoing based on larger cancer cohort with different controls.

In Figure 5d, the red lines indicated the peaks that were only present in more than 20% of patients with the six cancer types, but not in HC, patients with BT and patients with INF. Therefore, these new peaks represented potential cancer-specific alterations. Furthermore, we also observed the changes of peaks to some extent in the ccf-mtDNA fragmentation profile of patients with INF/BT compared to those of healthy controls. One possible reason is that certain inflammation diseases such as hepatitis or liver cirrhosis may exert a systemic influence on the chromatin conformation of the mitochondrial genome. Another

possible reason is that the disease state of INF/BT can lead to changes in the type or level of nucleases, which in turn leads to different fragmentomic features. We have provided related explanation for this point in the Discussion section on Page 20-21 Lines 409-416.

Minor Comments:

-some grammatical errors throughout - Not the priority of revision and did not significantly impede my review of the manuscript, but the authors may wish to copy edit more thoroughly prior to publication if accepted.

Response: Based on reviewer's suggestion, grammatical corrections have been performed by Professional English language editing service. Certificate was attached.

editage

Editing Certificate

This document certifies that the manuscript listed below has been edited to ensure language and grammar accuracy and is error free in these aspects. The edit was performed by professional editors at Editage, a brand of Cactus Communications. The author's core research ideas were not altered in any way during the editing process. The quality of the edit has been guaranteed, with the assumption that our suggested changes have been accepted and the text has not been further altered without the knowledge of our editors.

MANUSCRIPT TITLE

Aberrant fragmentomic features of circulating cell-free mitochondrial DNA as novel biomarkers for multi-cancer detection

AUTHORS

Yang Liu^{1,2}, Fan Peng¹, Siyuan Wang¹, Huanmin Jiao, Miao Dang, Kaixiang Zhou, Wenjie Guo, Shanshan Guo, Huanqin Zhang, Wenjie Song, Jinliang Xing

ISSUED ON

June 27, 2024

JOB CODE

CVZQD_4

Prabh Grewal
Senior Vice President - Editage

editage helping you get published

Since 2000, Editage has helped over 430,000 authors publish around 3.2 million research papers in scholarly journals across over 1000 disciplines through editorial, translation, transcription, and publication support services. Editage is a brand of Cactus Communications (cactusglobal.com), a science communication and technology company.

GLOBAL :
+1(833) 979-0061 | request@editage.com

CHINA :
400-120-3020 或 021-6020-9400 |
rbiao@editage.cn

CACTUS.

-would be good to provide more details on processing pipeline for ATAC-seq data

Response: Based on reviewer's suggestion, we have added more details

on processing pipeline for ATAC-seq data in the Methods section on Page 29-30 Lines 601-608.

Referee #3 (Comments on Novelty/Model System for Author):

It's interesting to evaluate if deeper sequencing of mtDNA in cfDNA could be useful for cancer detection and tissue of origin, however I don't think this manuscript can be properly assessed without information on experimental design, such as batching scheme. Otherwise, it seems as though the performance of cancer detection could just be a batch effect.

Response: Based on the reviewer's suggestion, we have provided more technical information about experimental design in Method section on Page 26 Lines 522-530 as following: To minimize bias resulting from systematic errors in the experiment, we ensured that samples of different disease types are included in the same batch of experiments, and standardized experimental procedures were followed. Additionally, all PCR reactions were conducted within the central region of the PCR thermal cycler's reaction module. Notably, all experiments were carried out by two individuals in a centralized laboratory, utilizing the same instruments across all samples. Notably, the batching design of every sample included with clinical metadata at a sample level was shown in revised Table EV3.

Referee #3 (Remarks for Author):

"Furthermore, region-specific fragmentomic features of ccf-mtDNA were observed, which was associated with protein binding, base composition and special structure of mitochondrial DNA."

Using the public ATAC seq data to confirm this hypothesis, and there are some regions as shown in figure 2B that indicate this negative correlation, but on the whole, I don't see much relationship. Consider other approaches for visualizing the data to support this conclusion, such as a scatter plot of ATAC seq depth vs FSD score.

Response: Based on the reviewer's suggestion, we have added the scatter plot of ATAC seq depth vs. FSD score across the whole mitochondrial genome. As shown in revised Figure 2b and the Appendix Figure S9, an overall negative correlation was found between the standardized mtDNA depth and FSD score, with the Spearman correlation r of -0.3933 for whole mitochondrial genome. As expected, we observed very high negative correlations between the standardized mtDNA depth and FSD score in some regions and no correlations in another regions. In fact, the fragmentation of ccf-mtDNA is a very complex process, and we have only preliminarily demonstrated that it is associated with protein binding, base composition and special structure of mitochondrial DNA. Further studies are needed to identify previously unknown mechanisms of ccf-mtDNA fragmentation. We also described related information in the

Results section on Page 9 Lines 156-158 and Discussion section on Page 19 Lines 381-385.

While it's true that there is a modest difference in fragment length distribution by the four functional groups, it's not clear if the difference is in the expected direction. Further, it's not clear if this difference in fragmentation is biologically relevant.

Response: In fact, the difference in fragment length distribution by the four functional groups is in the expected direction and biologically relevant. Previous studies have shown that GC-rich nuclear chromatin has a more extended conformation than AT-rich chromatin, suggesting that GC-rich regions may be more susceptible to enzyme digestion, leading to shorter fragments (Dekker, 2007). By calculating the base composition of the four functional regions in the revised Cambridge mtDNA reference

sequence (rCRS), a significant difference in G&C content was observed among the D-loop, mRNA, rRNA, and tRNA regions, with the highest in the D-loop region (46.79%) and the lowest in the tRNA regions (36.63%). As expected, the distribution of ccf-mtDNA fragment size in the D-loop, rRNA, mRNA and tRNA regions ranked from short to long.

Furthermore, triple-stranded DNA may be the preferred substrate of the Endonuclease G, which is a nonspecific nuclease for all nucleic acid species and accounts for a large part of mitochondrial nuclease activity (Ohsato *et al*, 2002). As expected, the distribution of ccf-mtDNA fragment size in the D-loop, which is a triple-stranded region formed by the stable incorporation of a third short DNA strand, is the shortest than other regions. We hold the opinion that the difference in fragment length distribution by the four functional groups is related to the chromatin conformation, including protein binding, base composition, and the special structure of mtDNA.

"When comparing to non-cancer controls, six types of cancer patients exhibited aberrant fragmentomic features."

You claim that the analysis shows similar fragmentation profiles between HC, BT, and INF groups, however in 5B show that the distribution of correlation to median NC is significantly different between HC and BT and HC and INF. Suggest re-wording.

Response: Based on reviewer's suggestion, we have made related

revision in the Results section on Page 12 Lines 227-233.

Please clarify how mtDNA copy number is calculated.

Response: We have added related description on method of calculating mtDNA copy number in the Methods section on Page 30 Lines 611-615.

"Both internal and external validation cohorts demonstrated the excellent capacity of our model in distinguishing cancer patients from non-cancer control, with all area under curve higher than 0.9110. The overall accuracy of tissue-of-origin was 90.59% and 90.45% for six cancer types in two validation cohort, respectively."

Consider showing showing scores by stage and tissue type in a figure, as well as breaking out tables by stage I, II, etc.

Response: Based on the reviewer's suggestion, the performance of the MEFI score in distinguishing different cancer type (NSCLC, HCC, CRC, SOC, BC, and ccRCC) was evaluated separately. As shown in Appendix Figure S16 and Table EV8, the detection model achieved AUC of higher than 0.9277 in all cancer types. Furthermore, the MEFI score exhibited high performance in differentiating patients with MT at all stages from HC in both the internal and external validation cohorts. We have added related results in revised Appendix Figure S16 and Table EV8.

Tumor stage	Internal validation cohort		External validation cohort	
	n	Sensitivity (95%CI)	n	Sensitivity (95%CI)
NSCLC				
I	31	83.87% (67.37% - 92.91%)	46	86.96% (74.33% - 93.88%)
II	6	83.33% (43.65% - 99.15%)	6	100.00% (60.97% - 100.00%)
III	14	92.86% (68.53% - 99.63%)	20	100.00% (83.89% - 100.00%)
IV	9	100.00% (70.09% - 100.00%)	23	100.00% (85.69% - 100.00%)
HCC				

0	7	71.43% (35.89% - 94.92%)	7	85.71% (48.69% - 99.27%)
A	35	85.71% (70.62% - 93.74%)	50	98.00% (89.50% - 99.90%)
B	9	88.89% (56.50% - 99.43%)	22	100.00% (85.13% - 100.00%)
C	10	90.00% (59.58% - 99.49%)	18	100.00% (82.41% - 100.00%)
D	2	100.00% (17.77% - 100.00%)	3	100.00% (43.85% - 100.00%)
CRC				
I	7	100.00% (64.57% - 100.00%)	9	88.89% (56.5% - 99.43%)
II	16	100.00% (80.64% - 100.00%)	25	92.00% (75.03% - 98.58%)
III	11	100.00% (74.12% - 100.00%)	37	100.00% (90.59% - 100.00%)
IV	12	100.00% (75.75% - 100.00%)	16	100.00% (80.64% - 100.00%)
SOC				
I	15	86.67% (62.12% - 97.63%)	7	100.00% (64.57% - 100.00%)
II	4	75.00% (30.06% - 98.72%)	4	100.00% (51.01% - 100.00%)
III	22	95.45% (78.20% - 99.77%)	14	100.00% (78.47% - 100.00%)
IV	7	100.00% (64.57% - 100.00%)	9	100.00% (70.09% - 100.00%)
BC				
I	6	66.67% (30.00% - 94.08%)	5	80.00% (37.55% - 98.97%)
II	14	85.71% (60.06% - 97.46%)	13	84.62% (57.77% - 97.27%)
III	2	100.00% (17.77% - 100.00%)	1	100.00% (5.13% - 100.00%)
IV	3	100.00% (43.85% - 100.00%)	3	100.00% (43.85% - 100.00%)
ccRCC				
I	6	83.33% (43.65% - 99.15%)	8	75.00% (40.93% - 95.56%)
II	3	100.00% (43.85% - 100.00%)	7	85.71% (48.69% - 99.27%)
III	5	100.00% (56.55% - 100.00%)	4	100.00% (51.01% - 100.00%)
IV	2	100.00% (17.77% - 100.00%)	2	100.00% (17.77% - 100.00%)

Note: The staging classification of NSCLC, CRC, BC, and ccRCC relies on the TNM staging system developed by the American Joint Committee on Cancer (AJCC). The stratification of HCC follows the Barcelona Clinic Liver Cancer (BCLC) staging system. The staging classification of SOC is guided by the International Federation of Gynecology and Obstetrics (FIGO) guidelines.

"We also evaluated the accuracy of MEFI approach in an independent external validation cohort" it seems a stretch to call this an independent external validation cohort. Patients enrolled at the same time, in the same country, and presumably were processed with the assay at the same time.

Please clarify why this is an independent external validation cohort.

Response: We thank the reviewer for valuable comments. In our study, the patients in the training cohort and external validation cohort were respectively enrolled from Xijing and Tangdu Hospitals. Xijing

Hospital ranking top 10 across the whole country is much more famous than Tangdu Hospital. Therefore, cancer patients enrolled in Tangdu Hospital are most from local regions around Xi'an where are relative underdevelopment in economy, while those enrolled in Xijing Hospital are most from broad regions across the country, often with higher income. So, it is plausible to define those patients from Tangdu hospital as external validation cohort. Considering the reviewer's comments, just to be on the safe side, we removed "independent" in revised manuscript, which had no influence on our conclusion.

Please clarify the source of the healthy individuals. You mention that all donors enrolled from the 2 different hospitals: how did healthy individuals enroll through a hospital? Were these healthy staff volunteers, or some other source?

Response: In the present study, healthy controls were enrolled from individuals who underwent routine physical examinations and were still cancer free for at least 6 months of follow-up. We have added more details about the enrollment of healthy individuals in the Methods section on Page 23 Lines 461-463.

Please explain how samples are allocated to extraction, library, and sequencing batch. It's unclear if confounding by batching may be responsible for the high performance seen in cancer detection and/or tissue of origin.

Response: As reviewer stressed, batch effect is a major source of variation, which should be carefully considered in the fragmentomic analysis of ccf-mtDNA. In our study design, we have indeed tried our best to adequately control the variables which impact fragmentation profile, such as time to processing of samples, library batch, sample site, and PCR cycles. Therefore, to minimize potential bias resulting from systematic errors, samples of different disease types were included in the same batch of experiments, and standardized experimental procedures were followed. Additionally, all PCR reactions were conducted within the central region of the PCR thermal cycler's reaction module. Notably, the experiments were performed by utilizing the same instruments across all samples in a centralized laboratory. No obvious batch effect was found. Therefore, we have added more details about the experimental design in the Methods section on Page 26 Lines 522-530. And, we have added Table EV3 to indicate the sample site and batching design of every sample included with clinical metadata at a sample level.

Please clarify the experimental design of the technical replicates and provide further information on results. I was not able to follow the conclusion.

Response: Based on valuable suggestion of reviewer, we have added more details about the experimental design of the technical replicates in the Methods section on Page 25 Lines 502-511. In detail, to evaluate the

repeatability of our detecting procedure for plasma ccf-mtDNA fragmentomic features, 10 healthy individuals were randomly selected and each cfDNA sample was subject to three times of library construction and WGS. And no significant difference was observed in the detection of ccf-mtDNA fragmentomic features among the three replicates, which also suggested no notable bias among different test batches (all $P > 0.05$).

I looked at the linked GitHub repository, but repository defaulted to main which has no code. The master branch has the code instead. Please correct the branching so that it's obvious where to find the code. Further, the code for the random forest classifier, just shows fitting a random forest to a dataset, but does not demonstrate the described cross-validation. Please update to include the cross-validation. It would also be helpful to have the already processed features in a CSV in order to verify the findings.

Response: Based on the reviewer's suggestion, we have corrected the branching in the linked GitHub repository. Also, the related code for cross-validation has been updated in GitHub repository. The already processed features were included in a CSV for verification of our findings (https://github.com/Mitoomics/MEFI_code).

Multiple times, the authors say that Mathios 2021 performed tissue of origin classification, but this is inaccurate. Please correct.

Response: Corrected.

Please correct spelling of wild-type in figure 5F and 5G.

Response: We have made related correction in revised Figure 5f and 5g.

References

- Bao H, Wang Z, Ma X, Guo W, Zhang X, Tang W, Chen X, Wang X, Chen Y, Mo S *et al* (2022) Letter to the editor: an ultra-sensitive assay using cell-free dna fragmentomics for multi-cancer early detection. *Mol Cancer* 21: 129. <https://doi.org/10.1186/s12943-022-01594-w>
- Cristiano S, Leal A, Phallen J, Fiksel J, Adleff V, Bruhm DC, Jensen SO, Medina JE, Hruban C, White JR *et al* (2019) Genome-wide cell-free dna fragmentation in patients with cancer. *Nature* 570: 385-389. <https://doi.org/10.1038/s41586-019-1272-6>

Dekker J (2007) Gc- and at-rich chromatin domains differ in conformation and histone modification status and are differentially modulated by rpd3p. *Genome Biol* 8: R116. <https://doi.org/10.1186/gb-2007-8-6-r116>

Dewitte A, Tanga A, Villeneuve J, Lepreux S, Ouattara A, Desmouliere A, Combe C, Ripoche J (2015) New frontiers for platelet cd154. *Exp Hematol Oncol* 4: 6. <https://doi.org/10.1186/s40164-015-0001-6>

Meddeb R, Pisareva E, Thierry AR (2019) Guidelines for the preanalytical conditions for analyzing circulating cell-free dna. *Clin Chem* 65: 623-633. <https://doi.org/10.1373/clinchem.2018.298323>

Ohsato T, Ishihara N, Muta T, Umeda S, Ikeda S, Mihara K, Hamasaki N, Kang D (2002) Mammalian mitochondrial endonuclease g. Digestion of r-loops and localization in intermembrane space. *Eur J Biochem* 269: 5765-5770. <https://doi.org/10.1046/j.1432-1033.2002.03238.x>

16th Sep 2024

Dear Prof. Xing,

Thank you for sending us your revised manuscript. We have now heard back from the two reviewers who agreed to re-evaluate your study. As you will see below, Reviewer #2 is generally satisfied with the revisions. Reviewer #3 noted that several key concerns regarding the validity and interpretation of the results remain insufficiently addressed and has indicated that the study is not suitable for publication in its current form. In principle, our editorial policy only allows a single round of major revision. However, considering the significance of the remaining issues and the clear suggestions provided by Reviewer #3, we have decided to give you the chance to address these remaining concerns, in an exceptional second round of major revision. Please note that the revised manuscript will be reviewed once again, so please make sure your response is as complete as possible.

On a more editorial level, please do the following :

- Please remove the Authors' contribution section from the manuscript file.
- Reference format: Please remove DOI links for published papers.
- Please rename "Competing interests" to "Disclosure statement and competing interests".
- Table EV3 should be renamed to Dataset EV1; the numbering of the subsequent tables will need to be adjusted accordingly.
- Appendix: please make sure that all files are compiled in one PDF; yellow highlights should be removed; page numbers should be added to the Table of Contents.
- Callouts: There is a callout for Suppl. Fig 1A, please correct the nomenclature; Please make sure the tables are called out in a sequential order.
- The revised figure legends should be added to the manuscript text file (without the figures).
- Data availability statement : please provide the specific URL for PRJCA020284 dataset. Please merge Code availability into Data availability section.
- Please expand the "results" section in "the paper explained".
- I have slightly modified the synopsis text(see attached). Please let me know if you are fine with it or if you would like to introduce further modifications.
- All Materials and Methods need to be described in the main text using our 'Structured Methods' format. According to this format, the Methods section includes a Reagents and Tools Table (listing key reagents, experimental models, software and relevant equipment and including their sources and relevant identifiers) followed by a Methods and Protocols section describing the methods, ideally using a step-by-step protocol format. The aim is to facilitate adoption of the methodologies across labs. Please download and fill our Reagents and Tools Table template (.docx), which you can find in our author guidelines: <https://www.embopress.org/page/journal/17574684/authorguide#structuredmethods>

An example of a paper with Structured Methods can be found here: <https://www.embopress.org/doi/10.15252/msb.20178071>.

- Figure legends:

1. Please note that the exact p values are not provided in the legends of figures 1b, d, f; 2b; 3a-e; 4a, c-h; 5b-c, e-f.
2. Please note that in figures 4a, c-d, f-h; 5b-c; there is a mismatch between the annotated p values in the figure legend and the annotated p values in the figure file that should be corrected.
3. Please note that the box plots need to be defined in terms of minima, maxima, and whiskers in the legends of figures 3a, e.

I look forward to seeing a revised form of your manuscript as soon as possible.

Use this link to login to the manuscript system and submit your revision: <https://embomolmed.msubmit.net/cgi-bin/main.plex>

Kind regards,
Jingyi

Jingyi Hou
Editor
EMBO Molecular Medicine

We require:

2) Individual production quality figure files as .eps, .tif, .jpg (one file per figure). For guidance, download the 'Figure Guide PDF': (<https://www.embopress.org/page/journal/17574684/authorguide#figureformat>).

3) A .docx formatted letter INCLUDING the reviewers' reports and your detailed point-by-point responses to their comments. As part of the EMBO Press transparent editorial process, the point-by-point response is part of the Review Process File (RPF), which will be published alongside your paper.

4) A complete author checklist, which you can download from our author guidelines (<https://www.embopress.org/page/journal/17574684/authorguide#submissionofrevisions>). Please insert information in the checklist that is also reflected in the manuscript. The completed author checklist will also be part of the RPF.

6) It is mandatory to include a 'Data Availability' section after the Materials and Methods. Before submitting your revision, primary datasets produced in this study need to be deposited in an appropriate public database, and the accession numbers and database listed under 'Data Availability'. Please remember to provide a reviewer password if the datasets are not yet public (see <https://www.embopress.org/page/journal/17574684/authorguide#dataavailability>).

- For the figures that you do NOT wish to display as Expanded View figures, they should be bundled together with their legends in a single PDF file called *Appendix*, which should start with a short Table of Content. Appendix figures should be referred to in

the main text as: "Appendix Figure S1, Appendix Figure S2" etc.

12) Author contributions: You will be asked to provide CRediT (Contributor Role Taxonomy) terms in the submission system. These replace a narrative author contribution section in the manuscript.

13) A Conflict of Interest statement should be provided in the main text.

14) Every published paper now includes a 'Synopsis' to further enhance discoverability. Synopses are displayed on the journal webpage and are freely accessible to all readers. They include a short stand first (maximum of 300 characters, including space) as well as 2-5 one-sentences bullet points that summarizes the paper. Please write the bullet points to summarize the key NEW findings. They should be designed to be complementary to the abstract - i.e. not repeat the same text. We encourage inclusion of key acronyms and quantitative information (maximum of 30 words / bullet point). Please use the passive voice. Please attach these in a separate file or send them by email, we will incorporate them accordingly.

15) Include a Reagents and Tools Table as part of the Methods section, which can be downloaded from our author guidelines (<https://www.embopress.org/page/journal/17574684/authorguide#structuredmethods>)

***** Reviewer's comments *****

Referee #2 (Remarks for Author):

I appreciate the author's clarifications in the text, particularly in the methods and the addition of Table EV3. The study is much more understandable now and I am reassured that the author's considered potential sources of technical variation in more detail. I do not wish to obstruct the publication of this paper as I understand that this journal's policy is not to consider second revisions - the results are interesting and the authors made a fair effort to respond to revisions.

Referee #3 (Remarks for Author):

Thank you for addressing some of the concerns raised in the previous review by providing additional data and clarifications. After consideration of the revised manuscript, supplementary materials, and updated GitHub repo, I still have reservations about the validity and interpretation of the results presented. My primary concerns are listed below:

1. Technical variation: despite the inclusion of donor-level annotation about batching, there remains insufficient information to properly assess the impact of technical variation on your results. The high performance of your model could potentially be

attributed to these confounding factors rather than true biological differences between cancer and non-cancer samples.

2. Pre-analytical variation: Upon reviewing the feature data provided in GitHub, I evaluated models that differentiated between various non-malignant groups (e.g., benign tumors, inflammatory conditions, and healthy controls) using the same features as the cancer detection analyses. I observed high discriminatory power. This observation raises concerns about the specificity of your classifier for cancer detection and suggests that factors unrelated to cancer biology may be driving the model's performance. It also makes me concerned that the process of donor enrollment, blood collection, or other types of pre-analytical variation are responsible for the high performance.

3. Generalizability: The exceptionally high performance metrics reported for cancer detection and tissue-of-origin classification are unusually good for liquid biopsy approaches. Without a more rigorous demonstration that these results are not due to confounding factors, it is difficult to accept their validity or potential for clinical translation.

Given these concerns, I do not believe the manuscript is suitable for publication in its current form. I recommend the following:

1. Consider additional analyses such as leave one batch out cross-validation, and demonstration that features that are important for cancer detection are robust across batches
2. Further information about donor enrollment and blood collection
3. Provide a detailed explanation of why the model shows high discriminatory power between non-malignant groups and how this impacts the interpretation of your results for cancer detection.
4. Discuss the limitations of your study more explicitly, particularly regarding the potential impact of confounding factors on the reported performance metrics.

Addressing these points would strengthen the manuscript and provide the necessary evidence to support your conclusions about the utility of ccf-mtDNA fragmentomics for cancer detection.

Referee #2 (Remarks for Author):

I appreciate the author's clarifications in the text, particularly in the methods and the addition of Table EV3. The study is much more understandable now and I am reassured that the author's considered potential sources of technical variation in more detail. I do not wish to obstruct the publication of this paper as I understand that this journal's policy is not to consider second revisions - the results are interesting and the authors made a fair effort to respond to revisions.

Response: Thank for the reviewer's comment.

Referee #3 (Remarks for Author):

Thank you for addressing some of the concerns raised in the previous review by providing additional data and clarifications. After consideration of the revised manuscript, supplementary materials, and updated GitHub repo, I still have reservations about the validity and interpretation of the results presented. My primary concerns are listed below:

1. Technical variation: despite the inclusion of donor-level annotation about batching, there remains insufficient information to properly assess the impact of technical variation on your results. The high performance of your model could potentially be attributed to these confounding factors rather than true biological differences between cancer and non-cancer samples.

2. Pre-analytical variation: Upon reviewing the feature data provided in GitHub, I evaluated models that differentiated between various non-malignant groups (e.g., benign tumors, inflammatory conditions, and healthy controls) using the same features as the cancer detection analyses. I observed high discriminatory power. This observation raises concerns about the specificity of your classifier for cancer detection and suggests that factors unrelated to cancer biology may be driving the model's performance. It also makes me concerned that the process of donor enrollment, blood collection, or other types of pre-analytical variation are responsible for the high performance.

3. Generalizability: The exceptionally high performance metrics reported for cancer detection and tissue-of-origin classification are unusually good for liquid biopsy approaches. Without a more rigorous demonstration that these results are not due to confounding factors, it is difficult to accept their validity or potential for clinical translation.

Response: Thank for your comment. We have added more analysis to demonstrate the robustness of our model using leave one batch out cross-validation. We also provided more description to explain why high discriminatory power was observed between various non-malignant groups. We also added more detailed discussion about the potential impact of confounding factors on the reported performance metrics. We really hope that this revision could eliminate your concerns.

Given these concerns, I do not believe the manuscript is suitable for publication in its current form. I recommend the following:

1. Consider additional analyses such as leave one batch out cross-validation, and demonstration that features that are important for cancer detection are robust across batches.

Response: Based on reviewer's suggestion, we have performed additional analyses in first three batches to further validate the robustness of our cancer detection models. The results were shown in revised Table EV 4. We also described related information in the Results section on Page 15 Lines 295-298 and Methods section on Page 33 Lines 677-679.

Table EV4 | MEFI performance for cancer detection using leave one batch out cross-validation.

Sequencing batch	Cohort	No. of Cancer	No. of Control	AUC (95%CI)
1	MT vs. HC	86	20	0.9994 (0.9975 - 1.0000)
	MT vs. BT	86	18	0.9567 (0.9071 - 1.0000)
	MT vs. INF	86	31	0.9672 (0.9407 - 0.9936)
2	MT vs. HC	75	27	0.9951 (0.9848 - 1.0000)
	MT vs. BT	75	16	0.9813 (0.9588 - 1.0000)
	MT vs. INF	75	34	0.9390 (0.8950 - 0.9830)
3	MT vs. HC	83	27	0.9987 (0.9956 - 1.0000)
	MT vs. BT	83	11	0.9677 (0.9259 - 1.0000)
	MT vs. INF	83	31	0.9627 (0.9322 - 0.9932)

Abbreviations: MT, malignant tumor; HC, healthy control; BT, benign tumors; INF, inflammation; AUC, area under the curve.

2. Further information about donor enrollment and blood collection.

Response: Based on reviewer's suggestion, we have added more information about donor enrollment and blood collection in the Methods section on Page 23 Lines 459-473 and Page 25 Lines 504-507 as following:

Cancer patients with pathological diagnosis were enrolled. Patients with MT receiving antitumor treatment prior to enrollment and those with

other types of concurrent tumors were excluded. The inclusion criteria for patients with BT or INF were as follows: (1) patients with BT or INF were diagnosed by pathological biopsy or patient's history, clinical manifestations and signs, laboratory test results, imaging findings, respectively; (2) still cancer free for at least 6 months of follow up. HCs were enrolled from individuals who underwent routine physical examinations based on inclusion criteria as follows: (1) with normal laboratory test results; (2) without a history of INF or BT within 5 years; (3) still cancer free for at least 6 months of follow up. All participants were untreated at the time of enrollment and had complete clinical data and demographic data, and provided the informed consent. The exclusion criteria for participants were as follows: (1) female participants who are pregnant or lactating or (2) with a history of cancer.

Sample collection was performed prior to treatment, and following the acquisition of diagnostic information. Whole blood samples (5 mL for each) were collected from patients with MT or BT 1 day before treatment and collected from patients with INF and HC at initial enrollment.

3. Provide a detailed explanation of why the model shows high discriminatory power between non-malignant groups and how this impacts the interpretation of your results for cancer detection.

Response: Based on reviewer's suggestion, we have provided a detailed explanation for this point in the Discussion section on Page 21

Lines 412-424 as following:

Furthermore, we also observed the obvious changes of ccf-mtDNA fragmentomic features in the patients with INF/BT compared to those of healthy controls, which underlies the discriminatory ability of our detection model between non-malignant groups. Possible explanations are that certain diseases such as hepatitis or liver cirrhosis may exert a systemic influence on the chromatin conformation of the mitochondrial genome and the type or level of nucleases in plasma or tissues, which in turn leads to different fragmentomic features among non-malignant groups or between non-malignant and malignant groups. Moreover, the difference among patients with INF/BT and HC may contribute to a heightened specificity in cancer detection by reducing false positives. Our data also highlights the need for a nuanced interpretation of our results.

4. Discuss the limitations of your study more explicitly, particularly regarding the potential impact of confounding factors on the reported performance metrics.

Response: Based on reviewer's suggestion, we have added related Discussion on limitations of our study on Page 22 Lines 445-450 as following:

Although our results presented here are highly promising, we have to acknowledge two major limitations which need to be addressed in future studies. One is about the potential impact of uncharacterized

confounding factors on the reported performance metrics. In addition, further evaluation of the MEFI approach is warranted in a larger cohort of patients with more types of cancer and even in a prospective cohort.

18th Oct 2024

Dear Prof. Xing,

Please find enclosed the final reports on your manuscript. We are pleased to inform you that your manuscript is accepted for publication and is now being sent to our publisher to be included in the next available issue of EMBO Molecular Medicine.

Kind regards,
Jingyi

Jingyi Hou
Editor
EMBO Molecular Medicine

Referee #3 (Remarks for Author):

I appreciate the work of the authors to add more information around potential impact of technical variation and donor enrollment. I think readers will appreciate seeing that batch effect did not influence the performance of the results. I also think that appropriate context has been added in the discussion around confounders influencing the performance, which should help with interpreting the results for clinical translation.
